# Structural Basis of the Interaction of the G Proteins, Gαi_1_, Gβ_1_γ_2_ and Gαi_1_β_1_γ_2_, with Membrane Microdomains and Their Relationship to Cell Localization and Activity

**DOI:** 10.3390/biomedicines11020557

**Published:** 2023-02-14

**Authors:** Rafael Álvarez, Pablo V. Escribá

**Affiliations:** 1Department of Biology, University of the Balearic Islands, 07122 Palma, Spain; 2Laminar Pharmaceuticals, 07122 Palma, Spain

**Keywords:** drug discovery, lipid rafts, membrane lipids, palmitoylation, protein structure, protein prenylation, lipid structure, protein–lipid interactions, membrane microdomain, cell signaling, membrane lipid therapy, melitherapy

## Abstract

GPCRs receive signals from diverse messengers and activate G proteins that regulate downstream signaling effectors. Efficient signaling is achieved through the organization of these proteins in membranes. Thus, protein–lipid interactions play a critical role in bringing G proteins together in specific membrane microdomains with signaling partners. Significantly, the molecular basis underlying the membrane distribution of each G protein isoform, fundamental to fully understanding subsequent cell signaling, remains largely unclear. We used model membranes with lipid composition resembling different membrane microdomains, and monomeric, dimeric and trimeric Gi proteins with or without single and multiple mutations to investigate the structural bases of G protein–membrane interactions. We demonstrated that cationic amino acids in the N-terminal region of the Gαi_1_ and C-terminal region of the Gγ_2_ subunit, as well as their myristoyl, palmitoyl and geranylgeranyl moieties, define the differential G protein form interactions with membranes containing different lipid classes (PC, PS, PE, SM, Cho) and the various microdomains they may form (Lo, Ld, PC bilayer, charged, etc.). These new findings in part explain the molecular basis underlying amphitropic protein translocation to membranes and localization to different membrane microdomains and the role of these interactions in cell signal propagation, pathophysiology and therapies targeted to lipid membranes.

## 1. Introduction

The development of medicines targeting cell membranes requires a deep knowledge of the molecular basis that rule lipid bilayer structure and how this structure regulates signaling protein interactions with membranes and the ensuing cell signaling. Moreover, these interactions could be altered in association with pathophysiological processes and can be modulated by interventions with bioactive lipids and synthetic derivatives with therapeutic activity (membrane lipid therapy, melitherapy). The peripheral signaling proteins, G proteins, are composed of one α, β and γ subunits. There are currently known to be 18 different human Gα subunits, 6 different Gβ and 12 different Gγ subunits [1,2,3,4]. Activation of G proteins by agonist-activated G protein-coupled receptors (GPCRs) provokes the dissociation of the Gαβγ heterotrimer into the Gα monomer and Gβγ dimer, each form regulating their respective effectors and other signaling proteins. In addition, Gα subunits can be modified at their N-terminus through the incorporation of myristic and/or palmitic acids, while γ subunits can be modified by the addition of farnesyl or geranylgeranyl moieties to their C-terminal cysteine; these modifications intervene in the interactions with membranes [1,5,6]. In this context, the abundant Gαi_1_β_1_γ_2_ complex, an adenylyl cyclase inhibitory (Gi) protein controlling cAMP levels, may contain one reversibly bound palmitoyl moiety as well as irreversibly bound myristoyl and geranylgeranyl moieties [7,8,9]. The aim of this study was to investigate the influences of membrane lipid structure and surface charge and the roles of G protein lipid modifications and charge on the interactions of G protein monomers (Gαi_1_), dimers (Gβ_1_γ_2_) and trimers (Gαi_1_β_1_γ_2_) with lipid membranes. The regulation of these protein–lipid interactions controls the localization of G proteins (and other amphitropic proteins) to different plasma membrane microdomains and organelles, where they may interact with distinct transmembrane and peripheral proteins, producing different signals [10]. The spatiotemporal organization of proteins forming signalosomes in defined membrane nano/microdomains is involved in many cellular processes [11,12]. Moreover, these signaling platforms formed in discrete membrane areas are relevant in pathological processes and their therapy [13].

The current view of the fluid mosaic model of the membrane [14] contemplates mosaicism in terms of a dynamically structured patchwork of membrane microdomains [15,16]. Under this new view of the fluid mosaic model, protein and lipid organization in the membrane follow a non-random co-distribution. Thus, proteins and lipids will form macromolecular structures or microdomains (small clusters of several nm or even μm) in conjunction with defined lipids and proteins in continuous dynamic turnover [15,17,18,19]. In this context, important changes in the membrane levels of one or more major lipids might be associated with the activation or deactivation of “lipid switches” that trigger or terminate crucial cellular processes, such as differentiation, cell proliferation, cell death, among others [20]. This fact further indicates the relevance of G protein–lipid interactions in human disease and therapy through approaches that control cell membrane lipid composition and structure. Lipid polymorphism (or lipid mesomorphism), the ability of lipids to form different stable or transient structures, is key to understanding how specific proteins interact with lipid bilayers and how these membrane microdomains regulate and are regulated by protein–lipid interactions [16,17,21,22,23]. Thus, lipids with a bulky polar head that are similar to cylinders (e.g., phosphatidylcholine—PC) give rise to lamellar structures, of which liquid ordered membrane microdomains (Lo) have been widely studied. Several classes of Lo membrane microdomains have been described, which are usually rich in sphingomyelin (SM) and cholesterol (Cho), and they have been termed *lipid rafts*. However, a variety of detergent-resistant microdomains may display specific features according to the presence and levels of caveolins, gangliosides, Cho, among others. By contrast, lipids with a small polar head (e.g., phosphatidylethanolamine [PE], diacylglycerol [DAG]) would be prone to form non-lamellar structures, such as inverted hexagonal phases (H_II_), and these are abundant in liquid disordered (Ld) lipid bilayer microdomains. The behavior of lipids in vitro reflects the membrane nano/microdomains that they form, which defines the location of important lipidated signaling proteins, such as G proteins [24]. The nature of the lipids covalently bound to these membrane proteins is also crucial. Thus, acylated proteins prefer raft-like liquid-ordered (Lo) domains while prenylated proteins are excluded from these membrane regions and prefer non-lamellar prone [24,25,26,27]. Moreover, the small G protein, KRAS, interacts preferably with membrane microdomains rich in negatively charged phospholipids (e.g., phosphatidylserine—PS), its isoprenyl moiety and positively charged C-terminal amino acids participating in the modulation of this binding with different PS nanodomains [28].

The movement of lipids and proteins in biological membranes is strongly regulated by the highly ordered *lipid rafts* [29,30]. Large unilamellar vesicles (LUVs) composed of equimolar quantities of PC, PE, Cho and SM can form raft-like microdomains with a high Cho and SM content, as in [31], wherein these were used as model rafts. PS is another important lipid, and it is predominantly localized to the inner monolayer of the plasma membrane [32] where it plays an important regulatory role [33,34,35]. Here, we demonstrated the existence of key interactions between the C-terminal polybasic domain of Gγ_2_ and PS in the binding of Gβ_1_γ_2_ and Gαi_1_β_1_γ_2_ to PS membranes. Moreover, we showed here that changes caused by reversible G protein palmitoylation influence the affinity for these charged microdomains. Interestingly, the present study reveals that G protein lipid anchors are crucial in the synergistic preference of heterodimers and heterotrimers with different lipid modifications for charged and non-lamellar prone membrane microdomains, which in part explains the affinity of peripheral protein peptides for PS-rich nanodomains [28]. In addition, each G protein subunit amino acid studied here plays a specific role in its binding to the lipid bilayer and later in its sorting to defined membrane microdomains. This extends previous data from our group [6,23,24,35,36], providing new relevant information about the precise influence of G protein and membrane lipid structures in G protein–membrane interactions that go beyond specific interactions of membrane proteins with defined lipids (e.g., ghrelin receptor and PIP2, [37]). Finally, the knowledge gained in this field can be used to design biomedicines to treat numerous conditions in which protein–lipid interactions are involved in pathophysiological cell signaling [38,39].

## 2. Materials and Methods

### 2.1. Materials

The pFastBac 1 vector was purchased from Invitrogen (Barcelona, Spain). Miller’s LB Broth culture medium and agarose D-1 were obtained from Conda Laboratories (Barcelona, Spain), while Grace’s medium was from GIBCO (Madrid, Spain). Penicillin and streptomycin were purchased from PAA (Pasching, Austria) and β-Mercaptoethanol from Acros Organics (Madrid, Spain). CHAPS was supplied by AppliChem (Darmstadt, Germany). The antibodies against Gαi_1_ (clone R4), Gβ_1_ (ref. sc-379) and Gγ_2_ (ref. sc-374) were obtained from Santa Cruz Biotechnology (Santa Cruz, CA, USA). IRDye 800CW-linked donkey anti-mouse IgG and IRDye 800CW-linked donkey anti-rabbit IgG were provided by Li-Cor Biosciences (Madrid, Spain). Palmitoyl-CoA, egg-PC, liver-PE, egg-SM, porcine brain-PS (phosphatidylserine) and Cho were obtained from Avanti Polar Lipids (Alabaster, AL, USA). The natural phospholipids used had a balanced proportion of saturated and unsaturated acyl chains, mainly palmitic, stearic, oleic, linoleic and arachidonic acids. Brain PS showed high levels of docosahexaenoic acid (DHA, 11%) and the main fatty acid found in egg SM was palmitic acid (86%). For further details, see the manufacturer’s website. GDP, GDPβS, HEPES, Tris-HCl, proteinase inhibitors and all other reagents were purchased from Sigma-Aldrich (Madrid, Spain).

### 2.2. Site-Directed Mutagenesis and Cloning of G Proteins

The cDNA encoding the recombinant Gαi_1_ protein was kindly provided in the pQE-60 expression vector (3.4 kb) by Prof. Alfred G. Gilman (University of Texas Southwestern Medical Center, Dallas). The cDNA encoding the human Gγ_2_ protein was generously provided by Dr Scott Gibson (Southwestern Medical School, University of Texas) and the amino acid sequence of this protein is shown in Table 1. Site-directed mutagenesis of the Gαi_1_ and Gγ_2_ proteins were carried out as described elsewhere [35], and in the case of the latter, it was performed using the primers shown in Table 2. The amino acid sequence and modifications of Gαi_1_ are shown elsewhere [35]. Finally, the cDNA encoding the human Gβ_1_ protein was a gift from Drs. Lutz and Niroomand (University of Heidelberg, Germany). The Gαi_1_, Gβ_1_ and Gγ_2_ cDNAs were cloned into the pFastBac 1 expression vector as described previously [35] and combined to produce the G protein monomers, dimers and trimers used in the present study (e.g., Appendix A).

### 2.3. G Protein Purification

#### 2.3.1. Gαi_1_ Proteins

Recombinant wild type and mutant Gαi_1_ proteins were overexpressed and purified using affinity chromatography as described previously [35]. Briefly, the recombinant proteins were expressed in Sf9 cells using the Bac-to-Bac Baculovirus Expression System (Invitrogen). Then, recombinant Gαi_1_ proteins were produced in Sf9 cells, which were cultured in suspension in Grace’s medium supplemented with 10% FCS (*v*/*v*), penicillin (100 units/mL) and streptomycin (100 μg/mL). The WT and Pal–Gαi_1_ subunits were purified from Sf9 cell membrane fractions after harvesting Sf9 cells by centrifugation and suspending them in 15 mL of ice-cold 20 mM HEPES buffer (pH 8.0) containing β-mercaptoethanol (10 mM), NaCl (100 mM), MgCl2 (1 mM), GDP (10 μM) and proteinase inhibitors (lysis buffer). The nuclei and unbroken cells were removed by centrifugation at 3000× *g* for 10 min at 4 °C, and the resulting sample was centrifuged again at 100,000× *g* for 1 h at 4 °C. The membranes recovered were suspended in 6 mL of HEPES buffer (50 mM, pH 8.0) containing β-mercaptoethanol (10 mM), NaCl (500 mM), CHAPS (16 mM), GDP (10 μM) and proteinase inhibitors, and after incubating for 1 h with gentle shaking, the membranes were centrifuged at 100,000× *g* for 1 h. The resulting supernatant (membrane extract) was dialyzed against HEPES buffer supplemented with GDP (0.5 μM) and leupeptin (50 ng/mL) and then purified by chromatography on a Ni-NTA column (1 mL of resin, Invitrogen). Subsequently, the resin was washed with 30 mL of HEPES buffer (20 mM, pH 8.0) containing β-mercaptoethanol (10 mM), NaCl (400 mM), C12E10 (0.05%, *v*/*v*), GDP (10 μM), leupeptin (0.5 μg/mL) and imidazole (15 mM), using an increasing and discontinuous thermal gradient (4, 17 and 25 °C). The column was then washed with 20 mL of HEPES buffer (20 mM, pH 8.0) containing MgCl2 (0.5 mM), β-mercaptoethanol (10 mM), NaCl (100 mM), C12E10 (0.05%, *v*/*v*), GDP (10 μM), leupeptin (0.5 μg/mL) and imidazole (15 mM) at 30 °C, and it was then activated with 10 mL of HEPES buffer (20 mM, pH 8.0) containing AlCl3 (30 μM), MgCl2 (50 mM) and NaF (10 mM, AMF buffer). Finally, the Gαi_1_ protein was eluted with HEPES buffer (20 mM, pH 8.0) containing β-mercaptoethanol (10 mM), NaCl (100 mM) and MgCl2 (1 mM, elution buffer) and was supplemented with a step gradient of imidazole (40, 80, 120, 240 and 300 mM). The purified protein was dialyzed and stored at −80 °C until use. The Myr–Gαi_1_ mutant protein was overexpressed and purified from the cytosolic fraction of infected Sf9 cells, which had been harvested and suspended in 5 mL of ice-cold lysis buffer as indicated above. The Myr–Gαi_1_ protein was purified from the supernatant by affinity chromatography as indicated above and fractionated by SDS-PAGE followed by coomassie blue staining.

#### 2.3.2. Gβ_1_γ_2_ Dimers

Gβ_1_γ_2_ complexes were purified as described previously [40] with some modifications. Briefly, Sf9 cells were co-infected with a recombinant baculovirus encoding wild type (WT) Gβ_1_ and Gγ_2_ (WT or mutant) subunits. Heterodimers were purified after harvesting Sf9 cells by centrifugation and resuspending them in ice-cold 50 mM HEPES buffer (pH 8.0) containing 10 mM β-mercaptoethanol, 500 mM NaCl, 10 μM GDP and proteinase inhibitors (lysis buffer). After cell nitrogen cavitation lysis at 500 p.s.i. for 30 min (Parr pump, [40]), nuclei and unbroken cells were removed by centrifugation at 3000× *g* for 10 min at 4 °C. Subsequently, 16 mM CHAPS was added to the sample and the resulting homogenate (10 mL) was shaken gently for 1 h at 4 °C and finally, it was centrifuged at 100,000× *g* for 1 h. The supernatant recovered was dialyzed against HEPES buffer containing 100 mM NaCl, 0.5 μM GDP and 50 ng/mL leupeptin, and the GDP concentration was then increased to 1 mM before the Gβ_1_γ_2_ dimers were purified by chromatography on a Ni-NTA column (0.5 mL of resin with a maximum binding capacity of 2.5–5 mg protein: Invitrogen). The resin was washed with 20 mL of 20 mM HEPES buffer (pH 8.0) containing 10 mM β-mercaptoethanol, 400 mM NaCl, 0.05% C12E10 (*v*/*v*), 1 mM GDP and 0.5 μg/mL leupeptin, following an increasing and discontinuous thermal gradient (4, 17 and 25 °C). The column was then washed with 5 mL of 20 mM HEPES buffer (pH 8.0) containing 10 mM β-mercaptoethanol, 100 mM NaCl, 0.05% C12E10 (*v*/*v*), 1 mM GDP, 0.5 μg/mL leupeptin and 5 mM imidazole at 25 °C and with 20 mL of the same buffer at 30 °C. Gβγ dimers were eluted from the column using 15 mL of 20 mM HEPES buffer (pH 8.0) with 30 μM AlCl3, 50 mM MgCl2 and 10 mM NaF (AMF buffer), and the purified dimers were dialyzed and stored at −80 °C until use.

#### 2.3.3. Gαi_1_β_1_γ_2_ Heterotrimers

Purified Gαi_1_ monomers (WT, Pal– and Myr– Gαi_1_) and Gβ_1_γ_2_ dimers (WT and RKK Gβ_1_γ_2_) were combined in a 1:1.5 ratio (*w*:*w*), and the different samples were then lyophilized. The dry residue was suspended in water in the presence of 5 mM GDPβS and incubated at 30 °C for 30 min. The samples were then diluted in 20 mM HEPES buffer (pH 8.0) containing 0.4 mM DTT, 100 mM KCl, 0.05% C12E10 (*v*/*v*) and 0.5 mM GDP. The samples were shaken overnight at 4 °C. Each protein mixture was combined with an Ni-NTA resin and then centrifuged at 800× *g* for 2 min at 4 °C, and the pellets recovered were washed with 500 μL of 20 mM HEPES buffer (pH 8.0) containing 0.4 mM DTT, 100 mM KCl, 0.005% C12E10 (*v*/*v*) and 0.5 mM GDP. After four washes, the proteins to the affinity resin were eluted with 450 μL (3 × 150 μL) of 20 mM HEPES buffer (pH 8.0) containing 0.4 mM DTT, 100 mM KCl, 0.005% C12E10 (*v*/*v*), 0.5 mM GDP and 300 mM imidazole. The Gαi_1_β_1_γ_2_ complexes recovered were diluted in 20 mM HEPES buffer (pH 8.0) containing 1 mM DTT, 100 mM KCl, 1 mM EDTA and 50 μM GDP and desalted and concentrated using Amicon centrifugal filters with a molecular weight cut-off of 50 kDa (Millipore). Concentrated Gαi_1_β_1_γ_2_ heterotrimers were frozen in liquid nitrogen and stored at −80 °C until use (Appendix A).

### 2.4. Acylation Reaction in Gαi_1_β_1_γ_2_ Heterotrimers

Purified Gαi_1_β_1_γ_2_ heterotrimers containing a myristoylated WT Gαi_1_ protein were subjected to in vitro palmitoylation as described elsewhere [27,35] with some modifications. Briefly, heterotrimers (0.1–0.3 nmol) were incubated with 20 nmol of palmitoyl-CoA for 3 h at 30 °C in 1 mL of 20 mM HEPES buffer containing 2 mM MgCl2, 50 mM NaCl, 1 mM EDTA, 0.2 mM DTT, 0.5 μM GDP and 7.5 mM CHAPS (pH 7.6). The heterotrimers were then diluted in HEPES buffer (20 mM, pH 8.0), containing DTT (0.2 mM), KCl (100 mM), EDTA (1 mM) and GDP (50 μM) and concentrated using Amicon centrifugal filters of 50 kDa, as described above. Finally, the resulting complexes were stored at −80 °C until use.

### 2.5. G Protein Binding to Model Membranes

Model membranes (liposomes) were prepared as described elsewhere [35,36], and liposomes (1 mM) were incubated for 1 h at 25 °C with 150 ng of purified monomer, 100 ng of dimers or 50 ng of heterotrimers in a total volume of 300 μL. Unbound G proteins were then separated from the membrane-bound G proteins by centrifugation at 90,000× *g* for 1 h at 25 °C. Finally, membrane pellets were resuspended in 36 μL of 80 mM Tris-HCl buffer [pH 6.8], containing 4% SDS, and mixed with 4 μL of 10× electrophoresis loading buffer (120 mM Tris HCl (pH 6.8), 1.43 M β-mercaptoethanol, 2% SDS and 50% glycerol).

The binding of the different Gαi_1_ monomers and Gβ_1_γ_2_ dimers to membranes were performed and quantified similarly as described [35]. Briefly, proteins were submitted to electrophoresis on 10% polyacrylamide gels and then transferred to nitrocellulose membranes. In these assays, a 1:200 dilution of anti-Gβ_1_ was used to detect the Gβ_1_ protein in the dimers in both the pellet and supernatant fractions. The membranes were finally incubated at room temperature for 1 h with IRDye 800CW-linked donkey anti-mouse IgG (1:5000 diluted in blocking solution), and antibody binding was detected by near infrared fluorescence using an ODYSSEY near infrared radiation detection system (LI-COR Biosciences).

Binding of Gαi_1_β_1_γ_2_ heterotrimers to membranes was quantified as described elsewhere [24]. Briefly, the membranes were probed with the specific anti-Gαi_1_ (dilution 1:100) and anti-Gβ_1_ (dilution 1:200) antibodies in blocking solution, and the binding of these antibodies was detected with horseradish peroxidase-linked anti-mouse and anti-rabbit secondary antibodies, respectively. The signal was developed with the ECL Western blot detection system and ECL Hyperfilm (Amersham).

### 2.6. G Protein Structure Analysis

Protein sequences were obtained from the protein database record at the National Center for Biotechnology Information (http://www.ncbi.nlm.nih.gov accessed on 9 March 2010). The sequence identification numbers assigned by the International Nucleotide Sequence Database Collaboration to the different considered Gγ proteins are (Table 3): AAH29367.1 (Gγ_1_), AF493870.1 (Gγ_2_), AAH15563.1 (Gγ_3_), AAH22485.1 (Gγ_4_), AAH03563.1 (Gγ_5_), AAH53630.1 (Gγ_7_), AAH95514.1 (Gγ_8_), AAM12590.1 (Gγ_9_), AAH10384.1 (Gγ_10_), AAH09709 (Gγ_11_), AAM12593.1 (Gγ_12_) and AAH93760.1 (Gγ_13_). The secondary structure prediction of these Gγ proteins was performed on the Psi-Pred server using the default parameter settings. The sequences of these Gγ proteins were aligned with the CLUSTAL W (1.81) tool on the Biology WorkBench Interface (v. 3.2) as described elsewhere [35]. A bi-dimensional projection of the hypothetical C-terminal α helix of the Gγ_2_ protein was obtained with the HELIQUEST server, and in this projection, the one-letter code size was proportional to the amino acid volume.

The different human Gβ proteins were aligned using the CLUSTAL W (1.81) tool previously described, considering the Gβ proteins: AAM15918.1 (Gβ_1_), AAM15919.1 (Gβ_2_), AAM15920.1 (Gβ_3_), AAG18442.1 (Gβ_4_) and AAH13997.1 (Gβ_5_). The three-dimensional structure of the Gαi_1_β_1_γ_2_ heterotrimer (identifier number 49238 in the Molecular Modeling Database, MMDB) was displayed using the Cn3D 4.3 macromolecular structure viewer. Anionic and cationic amino acids in the trimer were detected and highlighted using this computer application.

Multiple alignment of all the known human Gγ proteins obtained with *CLUSTAL W (1.81)*. The last 23 amino acids of Gγ_2_ are highlighted, in particular, the C-terminal basic amino acids that are relevant to the protein–lipid interactions.

### 2.7. Data Analysis

The Origin software was used to analyze the data and perform the statistical analysis. Unless otherwise indicated, the results are expressed as the mean ± SEM values from at least three independent experiments. For G protein heterotrimers, protein–lipid interactions were measured using both anti-Gαi_1_ and anti-Gβ_1_ antibodies. Differences were considered statistically significant at *p* < 0.05. For the correlation analysis between the number of C atoms in the lipid anchors of amphitropic membrane proteins and the number of charged amino acids in their peptide-membrane interacting regions, a non-linear analysis with the seven groups of proteins indicated in Figure 10B was carried out using Origin to determine r, r^2^ and χ^2^.

## 3. Results

We have formerly investigated G protein–membrane interactions from different points of view and in the context of cell signaling, human pathophysiology and therapy [6,17,22,23,24,35,41,42,43,44]. The present study extends our previous investigations, providing a thorough report on the consequences of changes in relevant protein and lipid structural factors that drive the interaction of the three forms of G proteins, Gα monomers, Gβγ dimers and Gαβγ trimers, with model membranes that resemble different lipid microdomains. Thus, we have investigated the roles of both membrane lipid structure (phospholipid types, cholesterol content, surface charge, lateral pressure etc.) and G protein structure (lipidation and charge of the alpha and gamma subunits). In the present study, Myr- and Pal- mutants corresponded to the Gαi_1_ subunit with the N-terminal G2A and C3S mutations that prevent protein myristoylation (Myr-) or palmitoylation (Pal-), respectively. Pal+ corresponded in the present study to the wild type Gαi_1_ subunit, in which full protein palmitoylation was enzymatically achieved as described in the Materials and Methods Section 2. The basic C-terminal amino acids of the Gγ_2_ subunit, Arg-62, Lys-64 and Lys-65, were mutated to glycine (R62G, K64G and K65G, respectively) in single, double or triple mutants. In addition, the C68S mutation prevented the Gγ_2_ subunit geranylgeranylation and further processing (methylation and removal of the 3 C-terminal amino acids) [6], and it was studied in single or multiple mutants containing the above mutations. Finally, G protein heterotrimeric complexes of the wild type form and mutants of the alpha and/or gamma subunits (single or multiple combined mutations) were studied to define the role of these amino acids in the interactions with membranes. Altered protein–membrane interactions are associated with pathological processes and certain biomedicines, such as bioactive lipids, have been designed to treat relevant conditions.

### 3.1. Membranes Used in the Present Study

The plasma membrane is the preferred location for GPCR–G protein interactions. The most abundant phospholipid in this lipid bilayer is PC, which was used as a control to investigate the role of other membrane lipids on G protein–membrane interactions. *Lipid rafts* are L_o_ microdomains characterized by the abundance of SM and Cho, and they form spontaneously and segregate from L_d_ microdomains in PC:PE:SM:Cho (1:1:1:1, mole ratio) model membranes [45,46]. The intracellular leaflet of the plasma membrane has a high proportion of PE and PS, phospholipids involved in the formation of Ld (non-lamellar prone) and of negatively charged membrane microdomains, respectively. These lipids were combined with PC to establish a model of microdomains that can be used to study G protein associations at the inner membrane leaflet.

### 3.2. The Role of Gγ_2_ C-Terminal Region in Gβ_1_γ_2_–Membrane Interactions

Hitherto, 12 different human Gγ subunits have been identified to date. These Gγ proteins were aligned, and the key amino acids that interact with membrane lipids are indicated in Table 3. The WT human Gγ_2_ protein and 15 mutated forms of this G protein subunit were transfected and overexpressed in Sf9 cells (Appendix A and Table 4), and all the recombinant Gγ_2_ proteins produced were combined with the Gβ_1_ protein, purifying the Gβ_1_γ_2_ complexes formed by affinity chromatography on a Ni-NTA column (see ‘Materials and Methods’ Section 2 and Appendix A). Single and multiple point mutations of the Arg-62, Lys-64, Lys-65 and Cys-68 amino acids in Gγ_2_ were induced, and the mutant and WT Gβ_1_γ_2_ dimers were studied (Table 4). Significantly, this latter Cys-68 (C68S) mutation impeded the geranylgeranylation of this Gγ_2_ subunit. In addition to the lack of one isoprenyl moiety at the C-terminal region of the C68G Gγ_2_ mutant, the last three amino acids were not removed and the peptide was not methylated at the C terminus [6]; all these alterations caused important changes in the physicochemical properties of the Gγ_2_ subunit site for membrane interactions (polarity and hydrophobicity). This may account for its different electrophoretic mobility compared to the wild type Gγ_2_ subunit. When all these dimers were analyzed independently and immunodetected with anti-Gβ_1_ and anti-Gγ_2_ antibodies, the geranylgeranylated heterodimers had a similar electrophoretic mobility on non-denaturing polyacrylamide gels, which clearly differed from those of the non-geranylgeranylated counterparts (Appendix A).

The sequences of wild type Gγ_2_ and the mutants generated to include all the possible combinations of mutations of the key amino acids are shown. Key amino acids are shown in blue and the corresponding mutants are shown in black italic case.

#### 3.2.1. Geranylgeranyl Is Critical for the Membrane Binding of Gβ_1_γ_2_

Gβ_1_γ_2_–membrane interaction assays were performed, in which the aforementioned WT and mutant Gβ_1_γ_2_ proteins were combined with model membranes that mimicked several membrane microdomains. The most outstanding effect was witnessed with the Gβγ C68S mutant. The membrane binding of the dimer that carried this mutation was drastically reduced for all the model membranes used in this study compared to those of isoprenylated Gβ_1_γ_2_ dimers (Figure 1).

#### 3.2.2. Geranylgeranyl plus the Neighboring Basic Amino Acids Arg-62 and Lys-65 Drive Gβ_1_γ_2_ towards PE-Rich (Non-Lamellar Prone) L_d_ Membrane Microdomains

The interactions of geranylgeranylated Gβ_1_γ_2_ dimers with PE-rich model membranes (50% PE) were analyzed, with the K64G and R62G-K64G mutants showing a membrane binding profile similar to that of WT Gβ_1_γ_2_ (Figure 1E). Therefore, Lys-64 (K64G) did not play a relevant role in determining the preference of Gβ_1_γ_2_ for PE-rich microdomains (Figure 2A). In contrast, Arg-62 (R62G) and Lys-65 (R65G) were key amino acids in defining the preference of Gβ_1_γ_2_ for non-lamellar prone microdomains, although only Lys-65 was essential for this interaction (Figure 2). Hence, there was a relation between hydrophobicity in the Gγ_2_ C-terminal region, due to the introduction of serial mutations, and the decrease in the PC:PE-to-PC binding ratio (Figure 2B,C).

#### 3.2.3. Gγ_2_ C-Terminal Basic Amino Acids Drive the Interaction of Gβ_1_γ_2_ with PS-Rich Membranes

The ratios of PC-to-PC:PS (20% PS) binding showed that Lys-65 was the most relevant amino acid for these membrane–Gβγ interactions, followed by Arg-62 and Lys-64 (Figure 3A). The absence of Lys-65 in the Gγ_2_ subunit of the heterodimer practically abolished the Gβ_1_γ_2_ binding preference to PS-rich membranes (Figure 3B). However, Arg-62 and Lys-64 contributed similarly to this phenomenon, as indicated by the binding profiles of the R62G-K65G and K64G-K65G double mutants (Figure 3B). The triple mutation of the basic amino acids to glycine abrogated the differential interaction between Gβγ and membranes with and without PS (Figure 3B).

#### 3.2.4. Geranylgeranyl plus the Basic Arg-62, Lys-64 and Lys-65 Amino Acids Strongly Modulate the Interaction of Gβ_1_γ_2_ with PE- and PS-Rich Membranes

The Gβ_1_γ_2_ dimer exhibits a marked preference for membranes containing PE and PS, two major phospholipids with an asymmetrically higher distribution at the inner leaflet of the plasma membrane. The Gγ_2_ C-terminal region appears to be critical for the interaction of Gβ_1_γ_2_ heterodimer with membranes. Thus, single or multiple point mutations of R62G, K64G and K65G, in conjunction with the C68S mutation in the gamma subunit, reduced the affinity for PE- and PS-rich microdomains (Figure 3C). The remarkable differences in WT Gβ_1_γ_2_ binding to PC, PE and PS membranes contrasted with the relatively constant binding of the R62G-K64G-K65G Gβ_1_γ_2_ mutant to each of these different model membranes (Figure 3D). Hence, the Gγ_2_ C-terminal basic amino acids appear to be the key drivers of the preference of Gβ_1_γ_2_ for PE- and PS-rich microdomains (Figure 2C and Figure 3D).

The presence of the two major phospholipids in the inner membrane monolayer produced a synergistic effect on Gβγ binding to membranes. This effect was dampened severely by the K64 mutation, and it was completely abolished by the R62G-K64G mutations (Figure 3E). Finally, heterodimers carrying the R62G, K64G-K65G, R62G-K65G and K65G mutations had a low affinity for PE-rich membranes, and they formed a group with very similar binding affinities to PC:PS and PC:PE:PS membranes (Figure 3C,F). These data clearly show co-operation between the geranylgeranyl moiety and its neighboring basic amino acids in terms of the interaction of the Gβ_1_γ_2_ heterodimer with model membrane microdomains containing PE and PS.

#### 3.2.5. Arg-62 and Lys-65 Are Critical Residues in the Interaction between Gβ_1_γ_2_ and Ordered Lamellar Membranes

The Gβγ dimer prefers disordered (Ld) membrane microdomains. The isoprenoid lipid (ger) and Lys-65 are responsible for the reduced binding preference to ordered lamellar (raft-like) membranes, although the former is critical to favor G protein binding to membranes. Thus, the ability of Gβ_1_γ_2_ to interact with Lo membranes increased remarkably in the absence of Lys-65 (K65G: Figure 3G). Arg-62 (R62G) mutation also reduced the heterodimer’s binding preference to lamellar membranes and its preference for Ld-like microdomains. Finally, both Lys-64 and Lys-65 (i.e., the K64G-K65G and R62G-K64G-K65G mutants) also drove the preference of the dimer for PC-rich membranes with respect to raft-like membranes (Figure 3G).

#### 3.2.6. Geranylgeranylation Drives the Localization of Gγ_2_ and Gβ_1_γ_2_ to Biological Membranes

In Sf9 cells, the Lys-64 mutation did not influence the binding of Gγ_2_ to membranes, while the absence of geranylgeranylation led to the localization of Gγ_2_ to the cytosol when overexpressed alone (Appendix A). Following its overexpression, WT Gβ_1_ accumulated in the membrane fraction (Appendix A), which could be explained by the possible aggregation of Gβ_1_ monomers due to the excess of this subunit relative to endogenous Gα and Gγ proteins. However, when both proteins were co-expressed, WT Gβ_1_ with Gγ_2_ C68S-K64G mutant further shifted the binding profile of WT Gβ_1_ 1 as well as that of Gγ_2_, which were both even more prominent in the cytosol (Appendix A).

### 3.3. Effects of the Gγ_2_ C-Terminal and Gαi_1_ N-Terminal Regions and Membrane Lipid Organization on Gαi_1_β_1_γ_2_-Membrane Interactions

In general, the binding of Gαi_1_β_1_γ_2_ to lipid bilayers with different composition was closer to that of the dimer and diverted from the membrane binding behavior of the monomer, although the acyl moieties on Gαi_1_ also induced modest modulation of heterotrimer–lipid interactions.

#### 3.3.1. Geranylgeranyl and Myristoyl Moieties Are Required for Gαi_1_β_1_γ_2_ Targeting to PE-Rich Non-Lamellar Prone Microdomains

There was a clear preference of geranylgeranylated and myristoylated Gαi_1_β_1_γ_2_ heterotrimers to bind to PE-rich (PC:PE, 1:1, mol:mol) membranes with a high non-lamellar propensity (Figure 4A). Thus, myristoylated and isoprenylated and non-palmitoylated (a naturally occurring trimer form) bound significantly more to PE-rich membranes than to PC membranes (Figure 4B). The absence of the myristoyl moiety in these complexes abolished their preference for non-lamellar prone membranes (Figure 4A,B), as evident when the heterotrimers were detected with different antibodies against the Gαi_1_ or Gβ_1_ subunits (Figure 4B). However, mutations in the Gγ_2_ polybasic domain did not significantly affect the interaction of the different trimers with PC:PE membranes (Figure 4).

#### 3.3.2. Gαi_1_ Myristoylation and Palmitoylation and the Gγ_2_ C-Terminal Polybasic Domain Regulate Gαi_1_β_1_γ_2_-PS Interactions

Geranylgeranylated and myristoylated (and non-palmitoylated) Gαi_1_β_1_γ_2_ heterotrimers bound significantly better to PC:PS membranes (PC: PS, 3:2, mol:mol) than to PC membranes (Figure 5). Triple mutation of the Gγ_2_ C-terminal basic amino acids in combination with palmitoylation of the Gαi_1_ subunit dramatically reduced the behavior of the geranylgeranylated and myristoylated Gαi_1_β_1_γ_2_ heterotrimer, particularly in terms of its interaction with PS-rich membranes (PC:PS, 3:2, mol:mol). In addition, the lack of the myristoyl moiety abolished the preference of Gαi_1_β_1_γ_2_ for PS-rich membranes. Again, similar results were obtained when the binding of heterotrimeric G proteins to model membranes was measured with both the anti-Gαi_1_ or anti-Gβ_1_ antibodies, supporting the appropriateness of this technique (Figure 5 and Appendix A).

#### 3.3.3. Myristoylated and Non-Palmitoylated Gαi_1_β_1_γ_2_ Complexes Have a High Affinity for PE- and PS-Rich Membrane Microdomains

The binding of myristoylated, geranylgeranylated and non-palmitoylated Gαi_1_β_1_γ_2_ heterotrimers to PC, PC:PE (PC:PE, 1:1, mol:mol) and PC:PE:PS (PC:PE:PS, 2:2:1, molar ratio) membranes were qualitatively similar to that observed for Gβ_1_γ_2_ (Figure 6A). This important result further explained and demonstrated the decisive role of Gβγ dimer in the localization of Gαiβγ heterotrimers to non-lamellar prone membrane microdomains [24]. None of the other mutated heterotrimers studied here displayed this behavior in regard to their interaction with PE- and PS-rich membranes (Figure 6B). Thus, the myristoylated, geranylgeranylated and non-palmitoylated heterotrimers preferentially bound to PC:PE membranes than to PC, and the presence of 20 mol% PS further enhanced the interaction of these complexes with membranes (Figure 6C,D). The triple Gγ_2_ mutant, the myristoylated complex containing the R62G-K64G-K65G mutations, interacted less with PC:PE:PS membranes (Figure 6E,F). In fact, this interaction was very similar to the interaction with PC, which suggests a decisive role of the three C-terminal basic amino acids of Gγ_2_ in the localization of Gαi_1_β_1_γ_2_ to membrane microdomains with a high non-lamellar propensity and a negative charge, such as those rich in PE and PS that are the most abundant phospholipids in the inner layer of the plasma membrane [32].

The two studied palmitoylated Gαi_1_β_1_γ_2_ heterotrimers had PC:PE:PS-to-PC and PC:PE-to-PC binding ratios that very close to the ratios of R62G-K64G-K65G Gβγ-Pal− Gαi_1_ (Figure 6B), and their corresponding binding profiles did not differ significantly, further supporting that both structural features (i.e., lipid moiety and Gγ_2_ polybasic domain) are relevant in the microdomain segregation of G protein trimers (Figure 6E,F).

Finally, the two studied non-myristoylated and geranylgeranylated heterotrimers did not show significant lipid binding preferences, with binding ratio values close to 1 (Figure 6B). Binding of these complexes to PC, PC:PE and PC:PE:PS membranes were very similar in all the cases, demonstrating that the mutants lost the ability of segregating to different membrane microdomains (Figure 6G,H).

#### 3.3.4. Gαi_1_ Myristic Acid and C-Terminal Gγ_2_ Basic Amino Acids Prevent Gαi_1_β_1_γ_2_ Targeting to Raft-like Membrane Domains

The non-palmitoylated WT Gαβγ protein displayed similar affinity for PC membranes and raft-like (PC:PE:SM:Cho) membranes, whereas the heterotrimer lacking myristic acid on the alpha subunit (WT Gβ_1_γ_2_-Myr^−^ Gαi_1_: G2A alpha subunit mutant) showed higher binding affinity for lamellar prone (PC) membranes. Conversely, the R62G-K64G-K65G Gβ_1_γ_2_-Myr^−^ Gαi_1_ complex displayed a clear preference for raft-like membranes (Figure 7A). In fact, this mutant showed higher binding affinity to raft-like membranes (PC:PE:Cho:SM, 1:1:1:1, mol ratio) than to PC membranes, a behavior that contrasts with that of wild type heterotrimers that apparently prefer to bind to non-raft PC membranes (Figure 7B).

### 3.4. The Gαi_1_ Monomer and the Corresponding Heterotrimer Differ Remarkably in Their Binding to Membranes

The Pal+ Gαi_1_ monomer bound more intensely to PC, PC:PS (3:2, mol:mol) and PC:PE:Cho:SM (1:1:1:1, mol:mol) membranes than the WT Gβ_1_γ_2_ -Pal+ Gαi_1_ heterotrimer (Figure 8, upper panel). In contrast, the binding of the WT Gβ_1_γ_2_ -Pal^−^ Gαi_1_ (C3S Gαi_1_ mutant) heterotrimer to non-lamellar prone membranes of PC:PE (1:1, mol:mol) and PC:PE:PS (2:2:1, mol:mol) was significantly higher than that of the Pal^−^ Gαi_1_ monomer (Figure 8, middle panel). In all cases, significantly more of the heterotrimeric Myr^−^ (G2A mutant) Gαi_1_ protein bound to model membranes than the corresponding monomer (Figure 8, lower panel). Heterotrimeric Myr^−^ Gαi_1_ also bound to biological membranes significantly more than the corresponding monomeric form. The Myr^−^ Gαi_1_ protein bound most intensely to biological (Sf9 cell) membranes influenced by Gβ_1_γ_2_ (37%), followed by Pal^−^ Gαi_1_ (23%) and finally by WT Gαi_1_ (18%) (Appendix A).

### 3.5. The Myristoyl and Geranylgeranyl Moieties Plus the Gγ_2_ C-Terminal Polybasic Domain Are Key Determinants of the Interaction of Gαi_1_β_1_γ_2_ with Biological Membranes

Myristic acid is essential in the interaction of G protein monomers or trimers with biological membranes. Thus, a mutation on the Gαi_1_ N-terminal glycine induced a significant loss of protein binding to Sf9 membranes relative to Pal^−^ (C3S) Gαi_1_ and WT Gαi_1_ (Appendix A). Nevertheless, the increase in the binding of the Myr^−^ (G2A) Gαi_1_ heterotrimeric form relative to the monomer was noteworthy, as indicated previously.

Geranylgeranylation had a more significant impact than myristoylation on the ability of the three subunits to interact with biological membranes. Thus, the ger^−^ K64G-C68S double Gγ_2_ mutant significantly modified the binding of Gαi_1_, Gβ_1_ and Gγ_2_ to cell membranes, unlike the K65G-C68S mutant (Appendix A). These results clearly demonstrate the importance of the C-terminal Gγ_2_ basic amino acids in the interactions of Gαi_1_β_1_γ_2_ with Sf9 cell membranes.

## 4. Discussion

Here, it was showed that the N-terminal region of Gαi_1_ (Myr- and Pal- mutants: G2A and C3S mutations) influence its interaction with membranes and that of the trimeric Gαi_1_β_1_γ_2_ protein. However, the membrane binding and microdomain sorting of the latter receive a greater influence from the C-terminal region of the Gγ_2_ subunit (C68S, R62G, K64G and K65G mutations), which has a membrane interaction pattern similar to that of the Gβ_1_γ_2_ dimer with slight modifications. In this context, the activity of G proteins is modulated significantly by protein–lipid interactions [24,35,36], and hence, we investigated the role of G protein structure and membrane composition on the membrane preference of monomeric, dimeric and trimeric forms of these proteins. To this end, model lipid membranes with a defined lipid composition were generated to test the binding of highly purified G proteins with specific mutations in the α and γ subunits. These new findings showing the role of lipid modifications and polybasic domain in G protein–membrane interactions, which support the previous suggestion about structure–function relationship on G protein interactions with lipids. These interactions, which remain largely unknown, are critical for cell signal propagation, and their alterations are involved in pathological processes and therapeutic approaches [47,48].

In this context, Gβ_1_γ_2_ dimer binding to negatively charged and non-lamellar prone lipid membranes apparently relies on the geranylgeranyl moiety and a neighboring Gγ_2_ polybasic domain that includes residues Arg-62, Lys-64 and Lys-65. In a previous study, we noted that the C-terminal region of the Gγ_2_ subunit regulated its subcellular localization [6]. Here, we found that geranylgeranyl and Lys-65 were minimal requirements to drive Gβ_1_γ_2_ to non-lamellar prone (PE-rich) microdomains. The binding of the Gβ_1_γ_2_ dimer to PE-rich domains was also regulated by Arg-62, and consequently, the R62G-K65G mutant was unable to bind preferentially to PE-rich non-lamellar prone membrane domains (Figure 9E). Conversely, the mutation of Lys-64 (K64G) did not dramatically affect the membrane binding of Gβ_1_γ_2_. In addition, all these basic amino acids participated in the direct binding of Gβ_1_γ_2_ to PS. Thus, the absence of the polybasic domain of Gγ_2_ completely abolished Gβ_1_γ_2_-binding to PS and its remarkable preference for PE- and PS-rich microdomains, the main phospholipids at the inner monolayer of the plasma membrane [32], and this indicates electrostatic protein–lipid interactions, as suggested by previous studies using confocal microscopy in live cells [6]. This study demonstrates that all the Gγ_2_ C-terminal modifications (isoprenylation, methylation and polybasic domain) have a differential role in G protein–lipid interactions.

Our results suggest that there is a specific structured interaction between the three basic amino acids studied and membrane lipids, in contrast with the data published that suggests the C-terminal Gγ_2_ region is disordered in the absence of membranes [49,50]. These previous studies give thorough and precise data about the overall α, β and γ subunit interactions, but the facts that the γ_2_ subunit had a C68S mutation that prevents its prenylations and no lipids were present in the medium could influence the structure elucidation of this small amino acid region. Current signaling models based on X-ray diffraction studies (Figure 10, [49]) consider the critical role of G protein lipids but not the exposure of charged amino acids to the membrane surface [51]. A structural prediction of the Gγ_2_ C-terminal region using PSIPRED confirmed previous X-ray diffraction results, since this simulation showed a disordered structure without regular spatial organization (‘random coil’). However, our results suggest the appearance of a transient structured conformational state for the C-terminal region of Gγ_2_ that interacts with the membrane lipid bilayer. In this model, confirmed by a bi-dimensional computer-assisted projection (Figure 9F), Arg-62 and Lys-65 would be very close to the polar membrane surface and near the geranylgeranyl moiety, whereas Lys-64 would be located at a distance in the turn of a small α-helix. This structure justifies the differential role of these basic amino acids in the interaction with negatively charged, PS-rich membranes.

Concerning Gαi_1_β_1_γ_2_ heterotrimers, the different acylated variants shared membrane lipid preferences with the Gβ_1_γ_2_ dimer, which differ from those of the Gαi_1_ monomer. These results further confirm and extend the pivotal role of the Gβγ dimer in Gαβγ–membrane interactions [24,35,36]. Thus, the preferential targeting of all the studied myristoylated and geranylgeranylated Gαi_1_β_1_γ_2_ complexes to PE-rich microdomains were observed. Therefore, myristoyl and geranylgeranyl moieties (irreversibly and permanently present in this G protein trimer) are key determinants of the binding of Gαi_1_β_1_γ_2_ to these non-lamellar prone membrane microdomains with a negative curvature strain, quite distinct from the more ordered membrane microdomains. Both these lipid modifications may be required to prevent Gαβγ localization into ordered lamellar membranes, such as lipid rafts. The high lipid packing density (i.e., high lateral surface pressure) in these membrane domains reduces the number of available gaps or “membrane defects” for the insertion of bulky lipids (e.g., geranylgeranyl). Conversely, the loose surface packing and membrane defects in non-lamellar prone membranes would facilitate the insertion of the myristoyl and geranylgeranyl moieties and even that of palmitic acid.

Previously, we demonstrated that Gβγ drives Gα from highly ordered lipid microdomains (L_o_) to disordered microdomains (L_d_) with a high curvature strain [24,35,36]. Here, we showed that palmitoylated Gαi_1_β_1_γ_2_ and de-palmitoylated Gαi_1_β_1_γ_2_ prefer non-lamellar prone model membranes, although the latter showed a higher binding capacity to membranes (Figure 9C). However, they differed significantly in their interaction with phosphatidylserine. Thus, while the myristoylated heterotrimer (WT Gβ_1_γ_2_-Pal^−^ Gαi_1_: C3S Gαi_1_ mutant) bound to PS-rich membranes (PC:PS and PC:PE:PS), palmitoylation disturbed the interaction of the heterotrimer with PS and even provoked electrostatic repulsion (Figure 5). Thus, the natural presence or absence of reversibly bound palmitoyl moiety regulated the interaction of Gαi_1_β_1_γ_2_ with membranes, as described for Gαi_1_ [35]. Interestingly, the palmitoylated heterotrimer that includes a Gγ_2_ subunit with mutations of Arg-62 (R62G), Lys-64 (K64G) and Lys-65 (K65G) had a similar binding profile to PS. This can be explained if the C-terminal basic amino acids of Gγ_2_ do not interact with membranes in palmitoylated Gαi_1_β_1_γ_2_. These results support the structure-based interaction of the C-terminal region of the gamma subunit with membranes and the role of the palmitoyl moiety in the regulation of protein structure and G protein–membrane interactions. Nevertheless, the Gβγ dimer again seems to be crucial in this interaction, more precisely, the Gβ_1_ protein (Figure 9B,D and Figure 10A).

Gβ_1_ is similar to a barrel with its widest side closest to the most hydrophobic lipidation points of the Gαβ®γ complex according to X-ray diffraction data [49]. On this wider face of the barrel, there is a notable presence of anionic amino acids (Figure 10A). Palmitoylation of Gαi_1_ may expose these negatively charged amino acids to the membrane surface while moving positively charged amino acids away from the membrane surface (Figure 9A,B and Figure 10A). In total, nine aspartates and three glutamates were identified in this region of the Gβ_1_ protein using the Cn3D 4.3 macromolecular structure viewer. Most of these anionic amino acids on the half of the G_β1_ closest to Gαi_1_ are totally conserved in all the human Gβ subunits (Figure 9A), and they might be the main residues responsible for the repulsion experienced by the palmitoylated heterotrimers. Interestingly, palmitoylation increases Gαi_1_ binding while it reduces Gαi_1β1_γ_2_ binding to biological and PS-rich membranes (Figure 8). Thus, in addition to their divergence in the affinity for nonlamellar regions, G protein monomer and heterotrimer also differ in their affinity for negatively charged membrane regions (Figure 5, Figure 6 and Figure 9A,B,D).

Finally, mutations in the basic Gγ_2_ C-terminal amino acids impede the binding of Gβ_1_γ_2_-Pal^−^ Gαi_1_ to PS, also demonstrating the important involvement of the Gγ_2_ polybasic domain in Gαiβγ–membrane interactions (Figure 5 and Figure 6). Mutations of Arg-62 (R62G), Lys-64 (K64G) and Lys-65 (K65G) impaired the preferential binding of WT Gβγ-Pal^−^ Gαi_1_ (C3S Gαi_1_ mutant) to PS membranes but not to non-lamellar prone membranes, consistent with earlier studies describing the prevalent role of lipid modifications in the targeting of Gαβγ to disordered membrane microdomains (L_d_ microdomains; [24,25,27,36]).

Proteins with one or more lipid modifications and their distinct levels of complexity are shown in Figure 10B. In total, 31 lipidated peripheral membrane proteins involved in cell differentiation and growth, synapsis or vesicle transport were analyzed. They were assigned to three different levels depending on the number of different lipid modifications. In this and previous works, we demonstrated the relevant role of one, two or three different lipid modifications of proteins that are involved in signal transduction [24,35,36]. We also analyzed the involvement of patches of basic amino acids in protein–membrane interactions combined with different lipid modifications and at different levels of complexity (Figure 10B). Our analysis showed that a limited number of palmitoylated proteins (15%) and none of the palmitoylated proteins at levels 2 and 3 interact with anionic lipids through their polybasic domains. Thus, palmitoylation appears to be a general regulator of the interaction between basic amino acids and anionic lipids (i.e., PS, PIPs). From a different point of view, myristoylated proteins have more basic amino acids in the membrane-interacting polybasic domain than palmitoylated and isoprenylated proteins (Figure 10B). Moreover, proteins with double or triple lipidation show less electrostatic interaction requirements than proteins with a unique lipid anchor. This observation is enormously relevant from a biological perspective considering the functional importance of the studied proteins, as it affects their membrane localization and microdomain sublocalization. In this context, we think that this work sheds light on the important modulatory role of the palmitoyl moiety and other membrane lipids on signal transduction and on other decisive events in cells. Thus, a nonlinear analysis showed an inverse correlation between the number of total C atoms present in the lipid anchors in the seven groups of lipid-modified proteins and the number of positively charged amino acids necessary for membrane binding (*n* = 7, r = 0.93; r^2^ = 0.87; χ^2^ = 1.54; Figure 10B). This indicates that although charged amino acids are required for the binding of proteins with no lipid anchors, in proteins with several lipid modifications, these amino acids are involved in membrane nanodomain localization rather than in the binding to membranes. Moreover, the higher the number of lipid modifications in an amphitropic protein, the higher its probability to bear a palmitoyl moiety (Figure 10), possibly due to the mobility among different microdomains that provides the reversible presence or absence of this lipid modification. However, further studies will be required to determine the precise roles of all these protein structures and membrane lipid compositions in protein–lipid interactions, cell signaling, pathophysiological processes and melitherapy.

Consequently, we propose here a different model for the interaction between G proteins and biological membranes based on the present and previous results [23,24,35]. According to this model, palmitoylated-myristoylated-geranylgeranylated Gαi_1_β_1_γ_2_ heterotrimers would interact with the corresponding GPCRs in anionic lipid-poor and PE-rich membrane microdomains with a high non-lamellar propensity (Figure 9D). This fully lipidated G protein heterotrimer is bound to the GPCR and activated upon ligand binding to the receptor. Then, the GDP associated with Gαi_1_ is exchanged for GTP and the doubly acylated (myristoylated and palmitoylated) Gαi_1_ protein dissociates from the Gβ_1_γ_2_ dimer. The turnover of the palmitic acid bound to Gαi_1_ is fast and consequently, the palmitoyl moiety is removed from its N-terminal end [52]. The non-palmitoylated Gαi_1_ protein has a higher affinity for PS-rich raft-like membrane microdomains, where it can inhibit adenylyl cyclase. The migration of the de-palmitoylated monomer toward microdomains with a high density of negative charge is dependent on the basic amino acids situated on the same side of its N-terminal α helix (Figure 9A; [35]).

All Gα proteins have GTPase activity and are inactivated by the hydrolysis of GTP to GDP. This reaction is promoted by GAPs (GTPase-activating proteins) and in this sense, the lipid environment of the GAP–Gα complexes may play a decisive regulator role. In fact, Gαi and RGS4 can interact with anionic lipids, and PS has been seen to influence RGS4 activity [35,53,54]. Furthermore, palmitoylation of Gαi_1_ inhibits its response to RGS4 [54]. In general, palmitoylation is a mechanism closely related to the Gα activation-deactivation cycle. Thus, palmitoylation of Gαi_1_ would induce a conformational switch on its N-terminal region, and rotation of the N-terminal α helix would move the basic amino acids away from the membrane surface. Consequently, negatively charged and hydrophobic amino acids on the opposite side of the helix would induce the migration of Gαi_1_ from PS-rich raft-like microdomains toward PS-poor raft-like microdomains (Figure 9).

By contrast, the free Gβ_1_γ_2_ heterodimer preferentially localizes to PE and PS-rich microdomains where the Gγ_2_ C-terminal basic amino acids Arg-62, Lys-64 and Lys-65 would interact directly with PS. A transient helicoidal structure will appear in this C-terminal region of Gγ_2_ upon Gαβγ activation and it appears to be essential for its mobilization to specific membrane microdomains (Figure 9F). Arg-62, Lys-65 and the isoprenoid lipid are responsible for the major preference of Gβ_1_γ_2_ for non-lamellar prone and negatively charged microdomains, where PS could interact with the three C-terminal basic amino acids of Gγ_2_. PS is likely to be found in these microdomains with high non-lamellar propensity, as demonstrated elsewhere [55,56,57]. The proximity isoprenyl moieties would induce co-operative binding of additional transducer molecules [16,36,43]. Interestingly, G protein isoprenyl and acyl moieties have different effects on membrane lipid structure, which demonstrates the role of membrane proteins in the regulation of the biophysical properties of membranes [36,43]. Moreover, membrane proteins also regulate the membrane (and therefore the cell) shape [58] Thus, two relevant lipids in cell signaling such as PE and PS can either be enriched in certain membrane micro/nanodomains or can co-segregate in biological membranes, where they strongly modulate the localization of signaling proteins [23,24,35,59,60,61]. Gβ_1_γ_2_ heterodimers would be targeted to these microdomains where effector proteins such as GRK2 may also be present. The new Gβ_1_γ_2_–GRK2 complex formed as a result would move to a non-lamellar prone GPCR domain where the receptor would be desensitized by GRK [24]. The inactive double acylated Gαi_1_ monomer would be localized to PS-poor raft-like microdomains, potentially signaling platforms where the inactive signal transduction machinery would be concentrated [62]. Thus, a receptor-Gβ_1_γ_2_–GRK complex and the palmitoylated Gαi_1_ monomer might converge in these raft-like domains where Gαi_1_ and Gβ_1_γ_2_ could re-associate to form a pre-active G protein heterotrimer and commence the G protein cycle again. Given the diversity of lipid modifications and polybasic domain amino acid sequences of the different G protein subunits known, it would be expected that their cellular localizations would be slightly different along their activity cycle and monomer/oligomer states and combinations. 

## 5. Conclusions

Here, we revealed important details about the basic mechanisms involved in the G protein–membrane interactions and how they influence the localization of the different physiological forms of these signal transducers: Gα monomers, Gβγ dimers and Gαβγ heterotrimers. Our study in part explains how the different G protein forms localize to different membrane microdomains, a critical step to establish the physical protein–protein interactions required for signal propagation [63]. The structural features of each G protein subunit (hydrophobicity, charged amino acids and their positions, lipid modifications), the membrane lipid composition and structure (surface charge, lipid order, nonlamellar propensity) and the reversibility of Gα protein acylation are crucial to orchestrate protein–lipid and protein–protein interactions. As GPCR signaling constitutes a major drug discovery target, the present study may be decisive for the rational design of new drugs [38]. Similarly, recent studies show that G protein-coupled receptors have predictable cholesterol binding sites without a consensus motif, which might have regulatory roles and be relevant for the development of therapies [64]. Indeed, this new knowledge has been used to develop new therapeutic strategies based on the use of synthetic lipids, termed membrane lipid therapy or melitherapy, some of which are currently undergoing clinical trials or are in the earlier stages of pharmaceutical development [38,44,47,65,66,67,68,69,70,71,72,73,74,75]. Finally, the present study in part explains the similarities and differences in the binding preference, kinetics and localization of different G protein subunits previously detected [76].

## Figures and Tables

**Figure 1 biomedicines-11-00557-f001:**
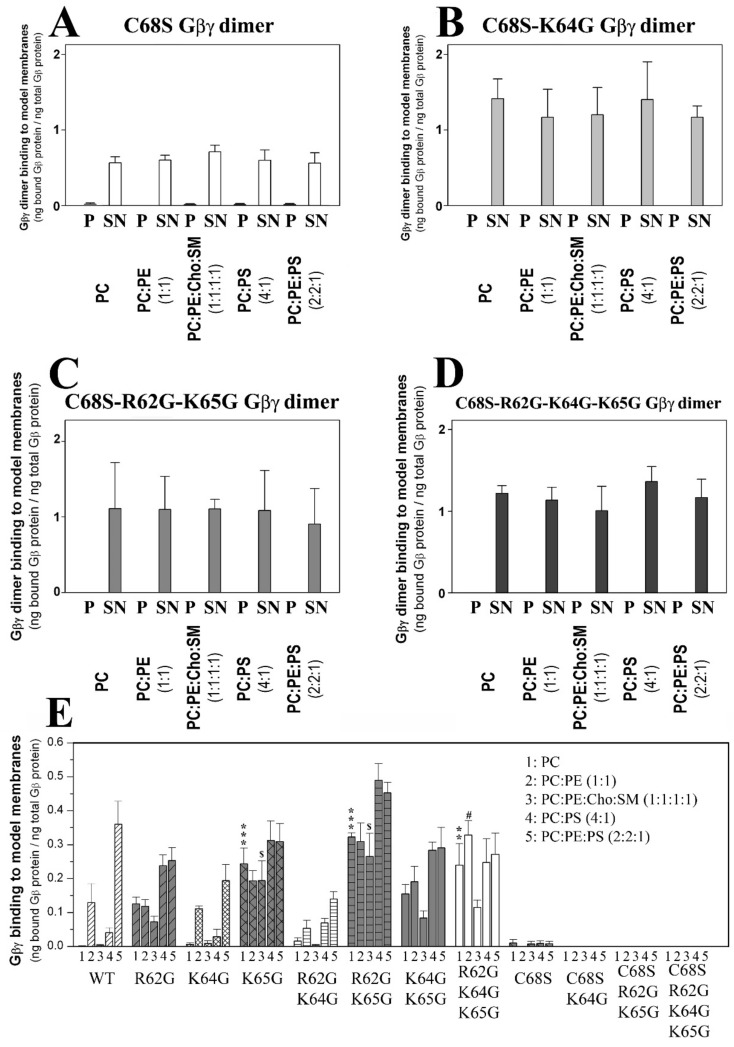
Effect of Gγ_2_ protein isoprenylation on Gβγ–membrane interactions. (**A**), Binding of non-geranylgeranylated Gβ_1_γ_2_ to different model membranes that are representative of biological membrane microdomains. The Cys-68 (C68S) mutation is responsible for impeding geranylgeranylation. (**B**), Binding of the non-geranylgeranylated Gβ_1_γ_2_ dimer carrying gamma-subunit mutations of Lys-64 (K64G mutant) and Cys-68 to model membranes. (**C**), Binding of the non-geranylgeranylated Gβ_1_γ_2_ dimer carrying triple mutations of Arg-62 (R62G), Lys-65 (K65G) and Cys-68 to model membranes. (**D**), Binding of Gβ_1_γ_2_ carrying mutations of Arg-62, Lys-64, Lys-65 and Cys-68 to model membranes. In all cases, the binding of the dimer is calculated as the ng of bound Gβ_1_ protein per ng of Gβ_1_ protein in the medium. The molar ratios used are indicated between parentheses. Representative immunoblots of each binding experiment are shown in the Appendix A section. The data represent the mean ± S.E.M values. (**E**), Gβ_1_γ_2_ dimer binding to model membranes. The graph shows the relative binding of the mutant and wild type (WT) dimers to different model membranes defined as ng of Gβ_1_ bound relative to the total ng of Gβ_1_ in the incubation medium. P, pellet (membrane fraction); SN, supernatant (soluble fraction). Symbols indicate significance (one, *p* < 0.05; two, *p* < 0.01; three, *p* < 0.001), where * corresponds to differences with respect to comparison with wild type G protein-PC interactions, # wild-type G protein-PC:PE interactions, and $ wild-type G protein-PC:PE:Cho:SM interactions.

**Figure 2 biomedicines-11-00557-f002:**
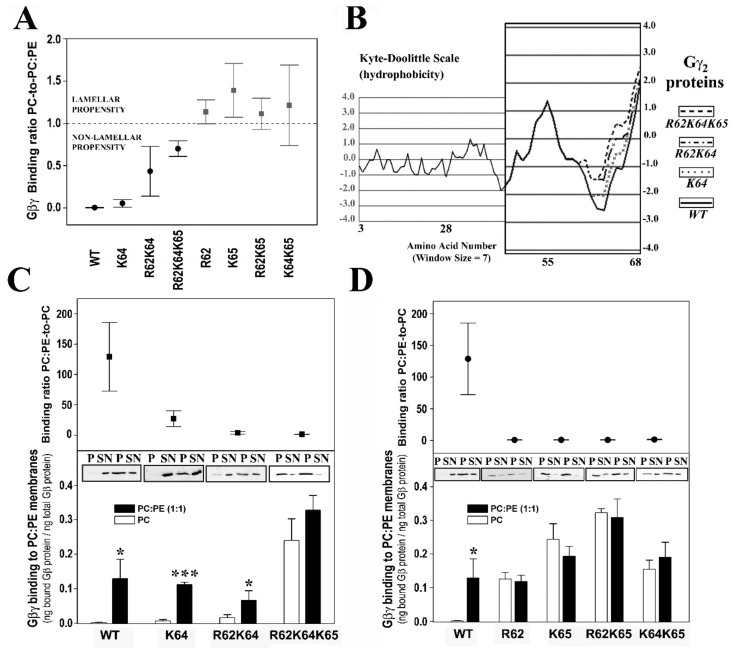
The influence of PE and Gγ_2_ structure on Gβγ-membrane interactions. (**A**), Binding ratio PC-to-PC:PE corresponding to the different geranylgeranylated dimers are generated. The ratios are calculated considering the fraction of Gβ_1_ bound to PC relative to the corresponding fraction bound to PC:PE (1:1 molar ratio) in each independent experiment. (**B**), Profiles of hydrophobicity of four representative geranylgeranylated Gβ_1_γ_2_ dimers on the Kyte-Doolittle scale. The increasing hydrophobicity of these dimers is correlated with their increasing ratio of PC-to-PC:PE binding. (**C**), Binding preference of Gβ_1_γ_2_ dimers for lamellar-(PC) and non-lamellar-(PC:PE) prone membranes. In the upper panel, the PC:PE-to-PC binding ratio is depicted (bound Gβ1 protein relative to total Gβi_1_ protein). (**D**), Binding of Gβ_1_γ_2_ dimer mutants that have no preference for non-lamellar prone membrane structures to PC and PC:PE membranes with respect to the wild type dimer (WT). The upper and lower panels are equivalent to the panels shown in (**C**). Representative immunoblots of each binding experiment are also shown in (**C**,**D**). In all cases, the data represent the mean ± S.E.M: *** *p* < 0.001; * *p* < 0.05. All binding ratios in panel A, except for K64G, are significantly different from the control (WT) PC-to-PC:PE binding ratio (*p* < 0.01). Labels: WT, wild type; R62, R62G; K64, K64G; K65, K65G; *p*, pellet; SN, supernatant.

**Figure 3 biomedicines-11-00557-f003:**
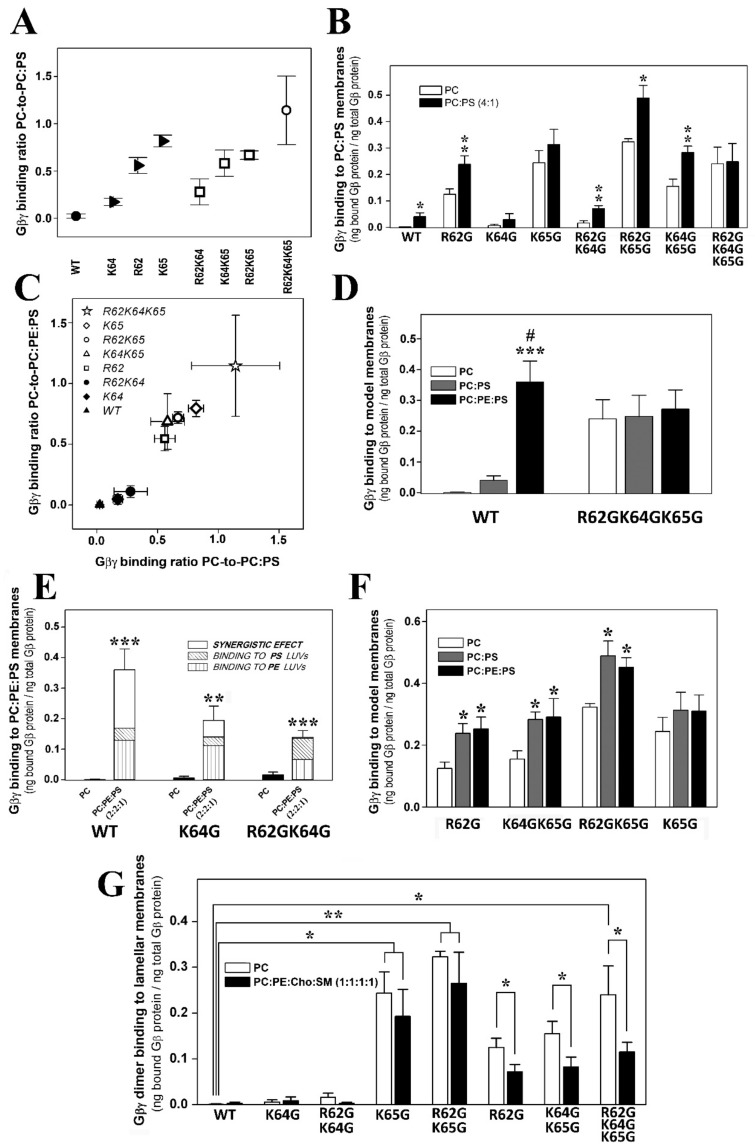
Roles of PS and Gγ_2_ structures on Gβγ–membrane interactions. (**A**), PC-to-PC:PS binding ratios corresponding to the different geranylgeranylated dimers. The wild-type dimer is depicted as a solid circle and the triple mutant as an open circle, whereas single mutants are depicted as solid triangles and the double mutants as open squares. (**B**), Binding of geranylgeranylated Gβ_1_γ_2_ dimers to PC and PC:PS (4:1, mol:mol) membranes. The binding of the dimer is calculated as the bound Gβi_1_ protein relative to the total Gβ_1_ protein (*n* = 3–8). Asterisks indicate significant differences (* *p* < 0.05; ** *p* < 0.01) with respect to PC membranes. (**C**), PC-to-PC:PE:PS binding ratio vs. the PC-to-PC:PS binding ratio. In this graph, the solid symbols represent the dimers with the greatest preference for PC:PE membranes, while the open symbols correspond to dimers with low or no preferential binding. (**D**), Binding of the WT and R62GK64GK65G mutant Gβ_1_γ_2_ dimers to PC, PC:PS and PC:PE:PS (2:2:1, mol:mol:mol) membranes. The bars represent the mean values of the dimer bound to these three model membranes (bound Gβ_1_ protein relative to total Gβ_1_ protein, *n* = 4). Significant differences with respect to PC (*** *p* < 0.001) and to PC:PS (# *p* < 0.05) membranes are found. (**E**), Binding of WT, K64G and R62GK64G Gβ_1_γ_2_ dimers to PC and PC:PE:PS membranes. Binding is calculated as in (**B**,**D**) (*n* = 3–4), and the bars corresponding to the total binding to PC:PE:PS membranes are divided into three parts: that representing the binding to PC:PE; that representing the binding to PC:PS; and a third part reflecting a possible synergistic effect. Significant differences (** *p* < 0.01; *** *p* < 0.001) with respect to PC membranes are found. (**F**), Binding of the R62G, K65G, K64GK65G and R62GK65G Gβ_1_γ_2_ dimers to PC, PC:PS and PC:PE:PS membranes. * *p* < 0.05 with respect to PC. (**G**), Binding of the WT, K64G, R62GK64G, K65G, R62GK65G, R62G, K64GK65G and R62GK64GK65G Gβ_1_γ_2_ dimers to PC (open bars) and PC:PE:Cho:SM (solid bars) membranes. The binding is calculated as in (**B**,**D**,**E**) (*n* = 3–8). The data represent the mean ± S.E.M.: *** *p* < 0.001; ** *p* < 0.01; * *p* < 0.05. ‘#’ indicates significant differences in the binding of Gβ_1_γ_2_ to PC:PE:PS with respect to its binding to PC:PS. All binding ratios in panels (**A**,**C**) are significantly different from the corresponding control (WT) PC-to-PC:PS and PC-to-PC:PE:PS binding ratios (*p* < 0.05).

**Figure 4 biomedicines-11-00557-f004:**
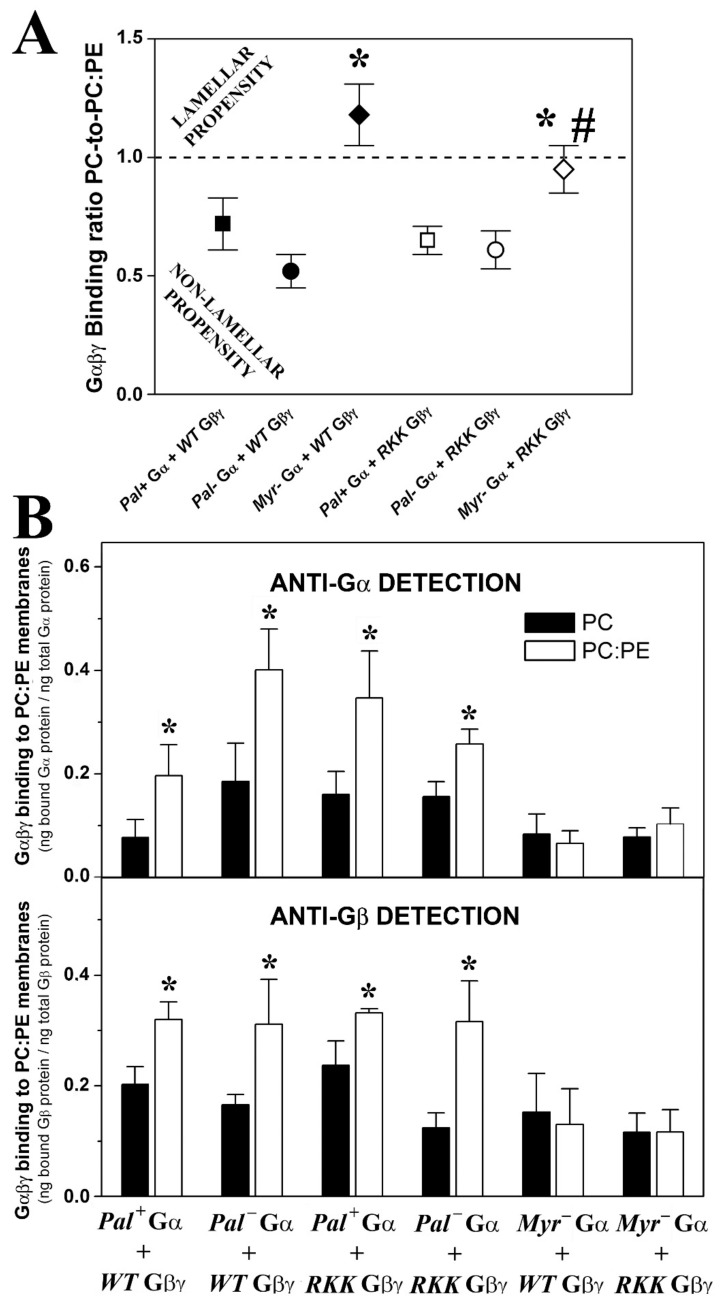
Effect of PE and Gαi_1_ and Gγ_2_ structures on Gαβγ–membrane interactions. (**A**), Binding ratio PC-to-PC:PE corresponding to the different Gαi_1_β_1_γ_2_ heterotrimers generated and purified. The ratios are calculated considering the fractions of Gαi_1_ and Gβ_1_ bound to PC relative to their corresponding fractions of binding to PC:PE (1:1, mol:mol) in each independent experiment. The data represent the mean ± S.E.M. values: * *p* < 0.05 with respect to Pal+ Gαi_1_/WT Gβ_1_γ_2_; # *p* < 0.05 with respect to Myr- Gαi_1_/WT Gβ_1_γ_2_. (**B**), Binding of Gαi_1_β_1_γ_2_ heterotrimers to PC and PC:PE membranes. Bars show the mean binding of the heterotrimers to PC and PC:PE membranes calculated as the bound Gαi_1_ protein relative to total Gαi_1_ protein (upper panel) or bound Gβ_1_ protein relative to total Gβ_1_ protein (lower panel). It has to be noticed that palmitoylation is reversible, so that Pal- is not a mutant but a natural status of the monomeric and trimeric G protein forms. RKK: R62G-K64G-K65G. The data represent the mean ± S.E.M.: * *p* < 0.05.

**Figure 5 biomedicines-11-00557-f005:**
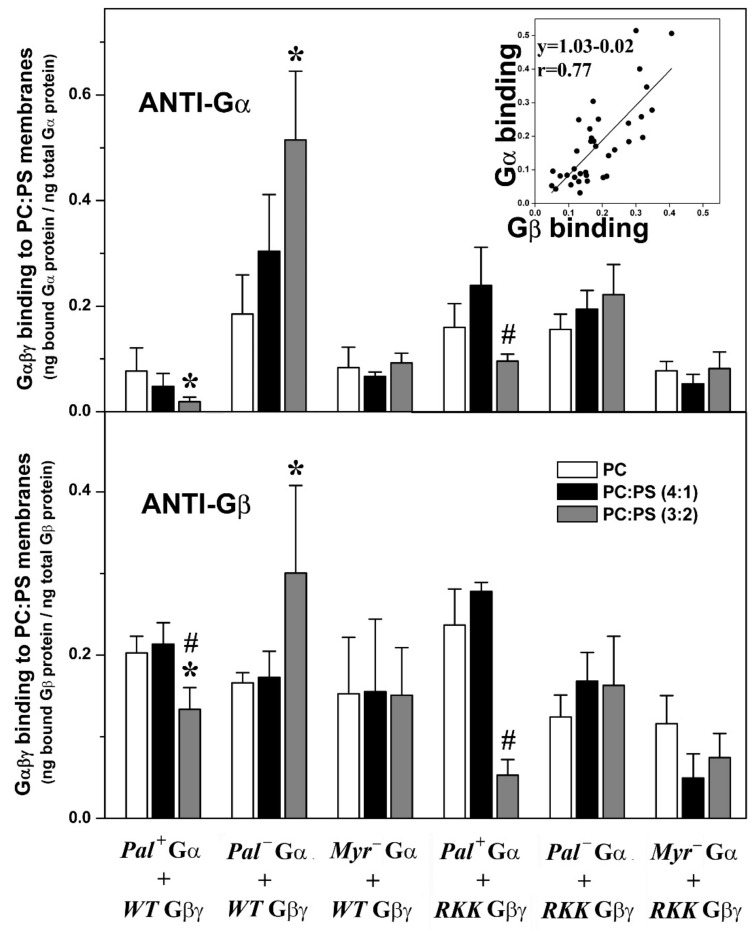
Effects of PS and the Gαi_1_ and Gγ_2_ structures on Gαβγ–membrane interactions. In the upper panel, the bars show mean ± S.E.M. values of Gαi_1_β_1_γ_2_ heterotrimer binding to PC and PC:PS (4:1 and 3:2, mol:mol) membranes, which is calculated as the bound Gαi_1_ protein relative to total Gαi_1_ protein. In the lower panel, binding is calculated as the bound Gβ_1_ protein relative to total Gβi_1_ protein. The inset shows the correlation between the binding for the G proteins investigated in the present study as measured with anti-Gαi_1_ and -Gβ_1_ antibodies using the same samples. The patterns of the bars in the lower panel are equivalent to those in the upper panel. RKK: R62G-K64G-K65G. Data correspond to 2–6 independent experiments, and “*” (*p* < 0.05) indicates significant differences with respect to PC membranes, whereas “#” (*p* < 0.05) indicates significant differences in the binding of Gαi_1_β_1_γ_2_ to PC:PS (3:2, mol:mol) with respect to its binding to PC:PS (4:1, mol:mol).

**Figure 6 biomedicines-11-00557-f006:**
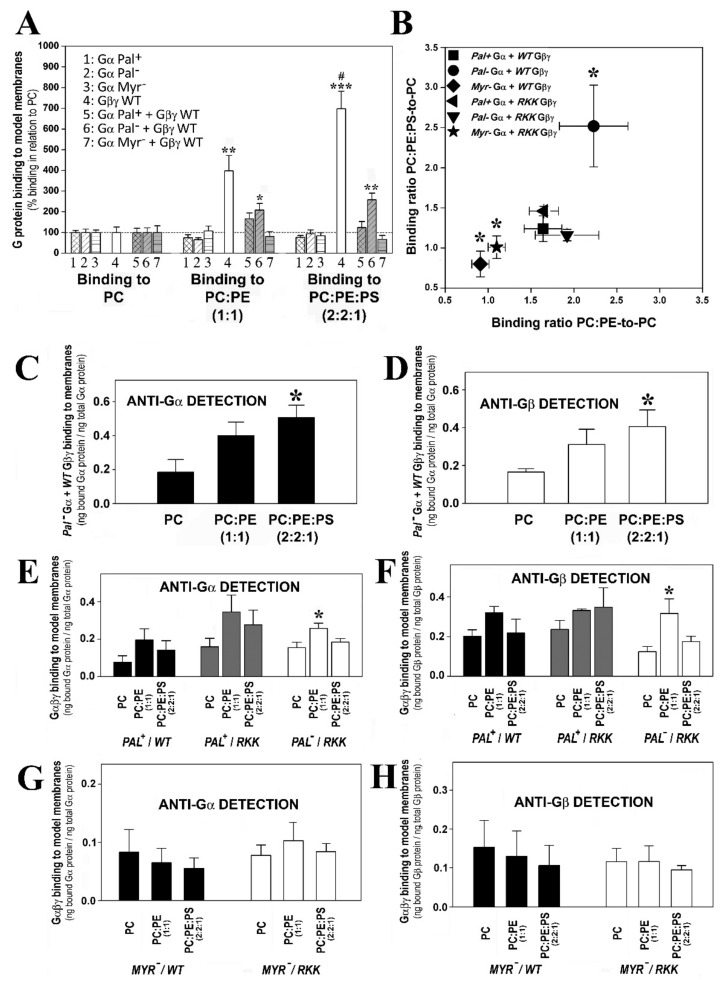
Effect of the major P-face membrane lipids, PS and PE and Gαi_1_/Gγ_2_ structure on the Gαβγ-membrane interactions. (**A**), Graph of the global G protein–membrane interactions. Binding of the G protein monomers, Pal+ Gαi_1_, Pal− Myr^−^ (G2A mutant) Gαi_1_ the WT Gβ_1_γ_2_ dimer, and the heterotrimers, WT Gβ_1_γ_2_-Pal+ Gαi_1_, WT Gβ_1_γ_2_-Pal^−^ (C3S mutant) Gαi_1_ and WT Gβ_1_γ_2_-Myr^−^ Gαi_1_, to PC, PC:PE (1:1, mol:mol) and PC:PE:PS (2:2:1, mol:mol). The bars represent the mean binding of G proteins to these three model membranes calculated as % binding relative to that to PC, considering the fraction of binding to PC as 100%. The asterisks indicate the significant differences in the binding of G proteins to membranes with respect to its binding to PC (*** *p* < 0.001; ** *p* < 0.01; * *p* < 0.05), while ‘#’ indicates significant differences (*p* < 0.05) in the binding of G proteins to PC:PE:PS with respect to its binding to PC:PE. (**B**), Binding ratio PC:PE:PS-to-PC vs. binding ratio PC:PE-to-PC. Each ratio is calculated considering the fraction of binding to PE membranes relative to the fraction of binding to PC in each independent experiment, and significant differences with respect to Pal+ Gαi_1_/WT Gβ_1_γ_2_ are indicated (* *p* < 0.01). RKK: R62G-K64G-K65G. (**C**,**D**), Binding of WT Gβ_1_γ_2_-Pal^−^ Gαi_1_ to PC, PC:PE and PC:PE:PS membranes. (**E**,**F**), Binding of WT Gβ_1_γ_2_-Pal+ Gαi_1_ (black bars), RKK Gβ_1_γ_2_-Pal+ Gαi_1_ (grey bars) and RKK Gβ_1_γ_2_-Pal^−^ Gαi_1_ (white bars) to PC, PC:PE and PC:PE:PS membranes. (**G**,**H**), Binding of WT Gβ_1_γ_2_-Myr^−^ Gαi_1_ (black bars) and RKK (R62G, K64G, K65G mutant) Gβ_1_γ_2_-Myr− Gαi_1_ (white bars) to PC, PC:PE and PC:PE:PS membranes. In (**C**,**E**,**G**), the bars represent the mean value of the α subunit binding to membranes (bound Gαi_1_ protein relative to total Gαi_1_ protein). In (**D**,**F**,**H**), the bars represent the mean value of the β subunit binding to membranes (bound Gβ_1_ protein relative to total Gβ_1_ protein). The data represent the mean ± S.E.M. values: * *p* < 0.05.

**Figure 7 biomedicines-11-00557-f007:**
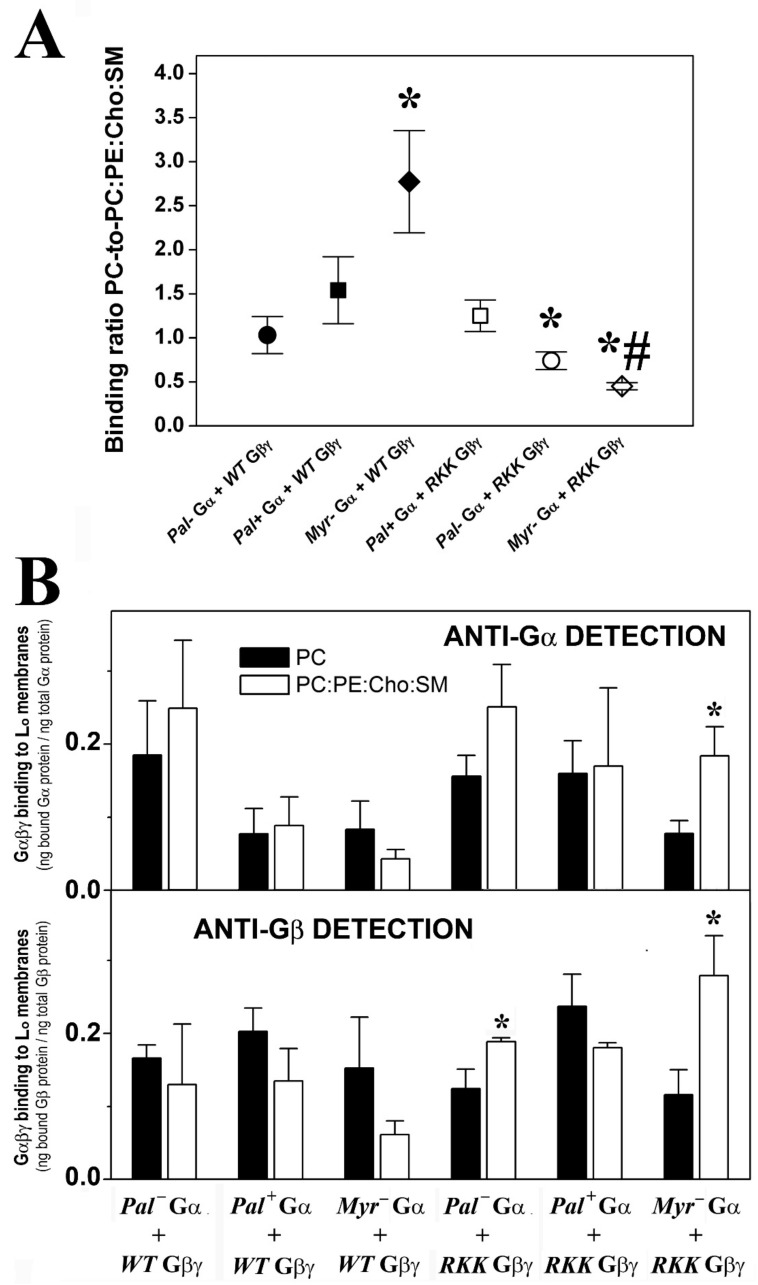
Effect of membrane lipid order and Gαi_1_/γ_2_ structure on Gαβγ–membrane interactions. (**A**), PC-to-PC:PE:Cho:SM binding ratio of the different Gαi_1_β_1_γ_2_ heterotrimer constructs. The ratios are calculated considering the respective Gαi_1_ and Gβ_1_ fractions bound to PC relative to their corresponding fractions bound to PC:PE:Cho:SM (1:1:1:1, mol:mol) in each independent experiment. The data represent the mean ± S.E.M. values: * *p* < 0.05 with respect to PC membranes; # *p* < 0.05. (**B**), Binding of Gαi_1_β_1_γ_2_ heterotrimers to PC and PC:PE:Cho:SM membranes. The bars in the upper panel show the mean binding of the heterotrimers to PC and PC:PE:Cho:SM membranes, calculated as the bound Gαi_1_ subunit relative to the total Gαi_1_ subunit. In the lower panel, the binding is calculated as the bound Gβ_1_ subunit relative to the total Gβ_1_ subunit. The colors of the bars in the lower panel are equivalent to those in the upper panel and in both panels representative immunoblots of each binding experiment are shown. The data represent the mean ± S.E.M.: * *p* < 0.05 with respect to PC membranes.

**Figure 8 biomedicines-11-00557-f008:**
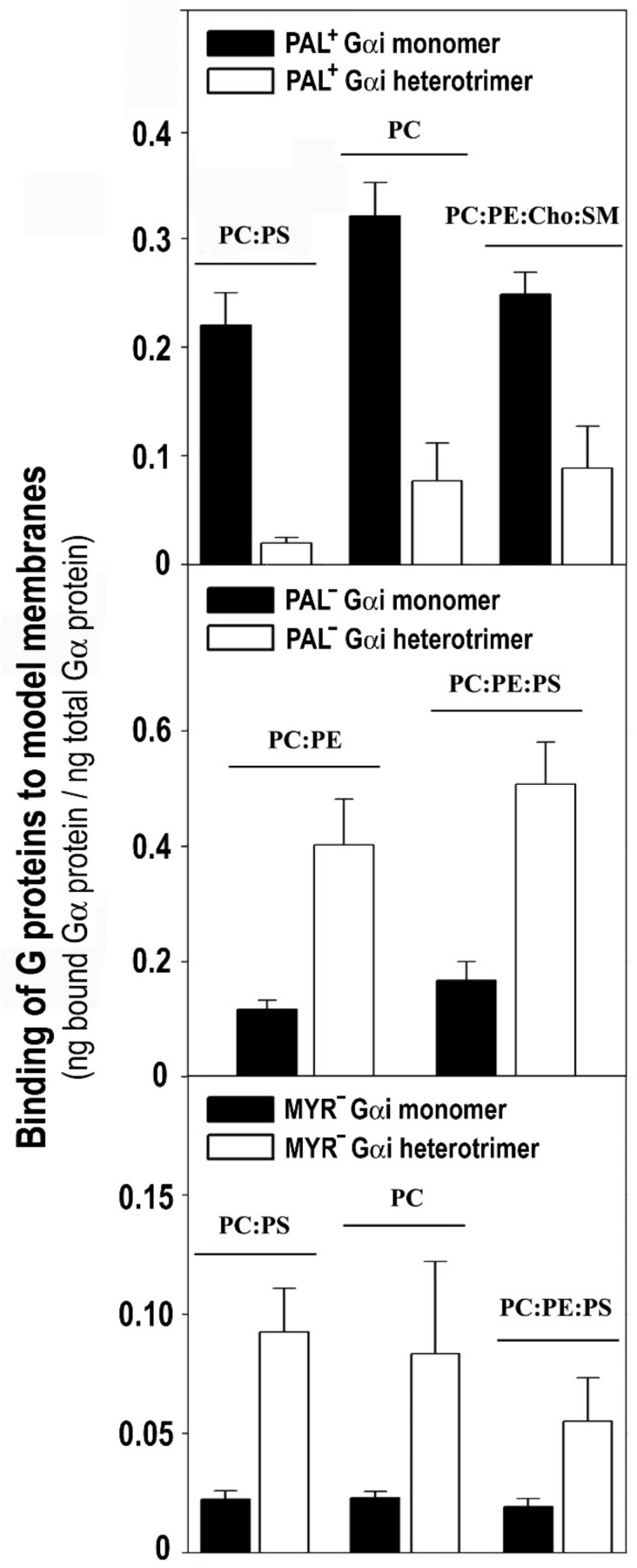
Comparative binding of Gαi_1_ monomer and Gαi_1_β_1_γ_2_ trimer to membranes with different lipid composition. This graph shows the most relevant differences in the membrane-binding properties of the Gαi_1_ protein in its monomeric and trimeric forms. Upper panel, Comparative binding of monomeric Pal^+^ Gαi_1_ and trimeric WT Gβ_1_γ_2_-Pal^+^ Gαi_1_ complex to PC:PS (3:2, mol:mol), PC and PC:PE:Cho:SM (1:1:1:1, mol:mol). The greatest differences in binding between these Pal^+^ Gαi_1_ proteins were observed in these cases. Middle panel, Comparative binding of monomeric Pal^−^ Gαi_1_ and trimeric *WT* Gβ_1_γ_2_-*Pal*^−^ Gαi_1_ heterotrimer to PC:PE (1:1, mol:mol) and PC:PE:PS (2:2:1, mol:mol). Lower panel, Comparative binding of monomeric Myr^−^ (G2A mutant) Gαi_1_ and trimeric WT Gβ_1_γ_2_-Myr^−^ Gαi_1_ complex to PC:PS (3:2, mol:mol), PC and PC:PE:PS (2:2:1, mol:mol). All bars are mean ± S.E.M. values and in all cases *p* < 0.05 for each monomer-trimer pair.

**Figure 9 biomedicines-11-00557-f009:**
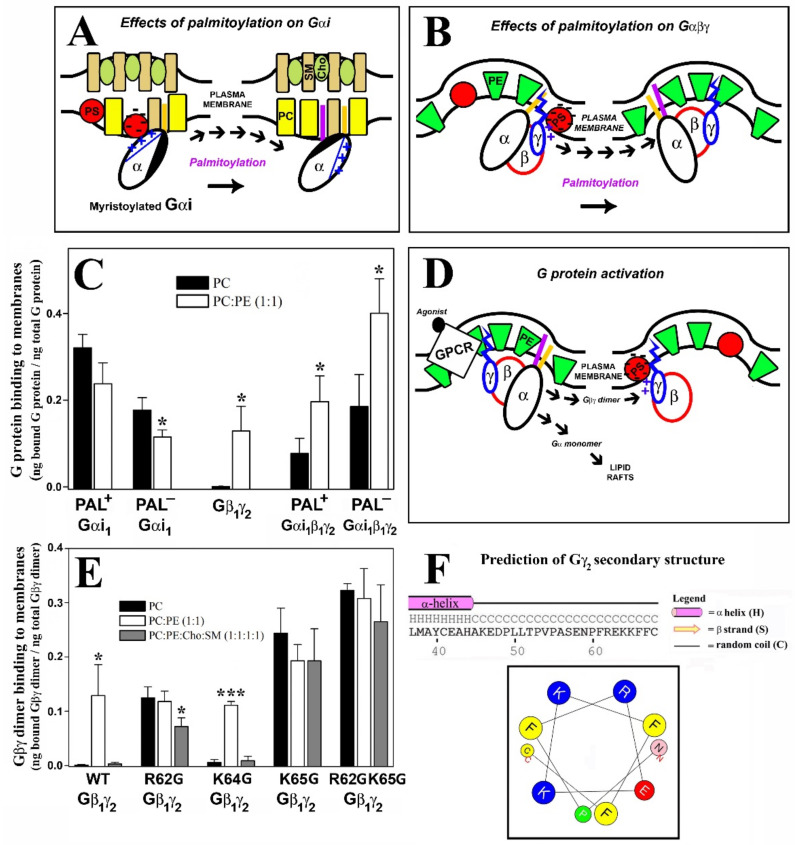
Key molecular determinants of Gαi1β1γ2-membrane interactions and models to explain the G protein-lipid membrane interactions. (**A**), Model to explain the interaction between Gαi1 and membrane lipids. A reorientation of the N-terminal α helix of Gαi1 by reversible palmitoylation of the protein drives its redistribution in the plasma membrane, moving it from PS-rich toward PS-poor raft-like microdomains. (**B**), Model to explain the Gαi1β1γ2-membrane interactions depending on the palmitoylation status of the protein G complex. Localization of palmitoylated Gαi1β1γ2 to PS-poor and PE-rich microdomains differs from the distribution of de-palmitoylated Gαi1β1γ2 in PS- and PE-rich microdomains. (**C**), Graph showing the nonlamellar preference and the role of reversible palmitoylation in the basic mechanisms of Gαi, Gβγ and Gαiβγ interactions with membrane lipids. Bars represent the binding of Pal+ Gαi1, Pal− (C3S mutant) Gαi1, WT Gβ1γ2, WT Gβ1γ2-Pal+ and WT Gβ1γ2-Pal− Gαi1 to PC and PC:PE (1:1, mol:mol) membranes. (**D**), Model of the activation of heterotrimeric G proteins that explains the interaction of Gαi1β1γ2 with membrane lipids in the presence of GPCR and the migration of the Gβ1γ2 dimer after its separation from Gα. (**E**), Graph showing representative results of the WT and mutated Gβ1γ2 dimer’s interactions with model membranes. Bars represent the binding of WT Gβ1γ2, single mutants and the R6G2K65G Gβ1γ2 mutant to PC, PC:PE (1:1, mol:mol) and PC:PE:Cho:SM (1:1:1:1, mol:mol) membranes to show the role of the γ2-subunit C terminal amino acids in the G protein dimer interactions with lamellar- and nonlamellar-prone membrane microdomains. (**F**), Secondary structure predictions of the C-terminal region of Gγ2 using the Psi-Pred and HELIQUEST tools. A bi-dimensional projection of the hypothetical C-terminal α helix of Gγ2 is depicted considering the last 10 amino acids of the protein (N and C represent the N- and C-terminal amino acids of this region, which coincidentally coincide with the amino acids asparagine, N, and cysteine, C, respectively). In (**C**,**E**), the data represent the mean ± S.E.M.: *** *p* < 0.001; * *p* < 0.05.

**Figure 10 biomedicines-11-00557-f010:**
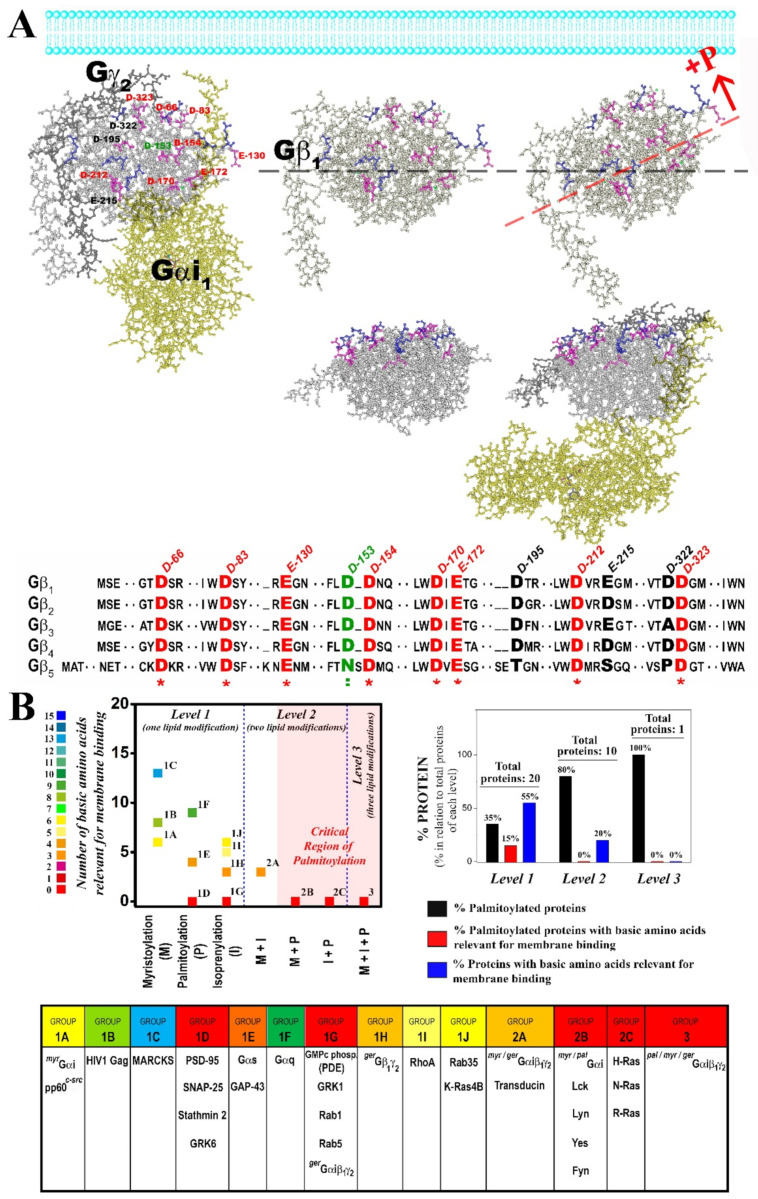
Lipid modifications to G proteins and other peripheral membrane proteins affect their interaction with anionic membrane lipids. (**A**), Possible orientation of Gαi1β1γ2 with respect to the surface of the plasma membrane. The incorporation of a palmitoyl moiety (+P) to Gαi1 would induce a conformational switch in the relative position of the Gαi1β1γ2 complex with respect to the lipid bilayer plane. Gβ1 would be tilted to the membrane side where the palmitic acid is bound to Gαi1 and consequently, anionic amino acids on the nearby Gβ1 half would approach membrane PS. These changes would drive the electrostatic repulsion experienced by the entire Gαβγ complex in the presence of PS. The anionic amino acids of this Gβ subunit that might be involved in this process are highlighted in a multiple alignment in the figure. Some of these amino acids are totally conserved and are shown in red. The structure shown corresponds to X-ray data coordinates provided by Wall et al. [49] and represented using the Molecular Modeling Database (see Materials and Methods). (**B**), Analysis of the existence of polybasic domains involved in the interaction with membrane lipids in functionally relevant proteins and their relation with the lipidation status of the proteins. Three different levels of complexity are established depending on the number of different lipid modifications the proteins are subjected to. Moreover, the number of basic amino acids involved in the interactions with anionic lipids are determined for each protein analyzed, showing that (a) palmitoylation and isoprenylation have lesser basic amino acid requirements and that (b) this requirement is even lesser in proteins with two or three lipid modifications. Finally, a graph shows the relation between protein palmitoylation and the presence of basic amino acids relevant for membrane binding.

**Table 1 biomedicines-11-00557-t001:** Amino acid sequence of the recombinant human Gγ_2_ protein.

H2N-MASNNTASIAQARKLVEQLKMEANIDRIKVSKAAADLMAYCEAHAKEDPLL TPVPASENPFREKKFFCAIL-COOH

**Table 2 biomedicines-11-00557-t002:** PCR primers used and their corresponding amino acid sequences.

Proteins	Forward Oligonucleotide
	5′-ATC**GAATTC**ATGGCCAGCAACAACACCGCCAGCATAGCACAAGCCAG-3′
	**Reverse Oligonucleotides**
*wild type* Gγ_2_	5′-CTC**GCGGCCGC**TTAAAGGATAGCACAGAA-3′
*GER-* Gγ_2_	5′-CTC**GCGGCCGC**TTAAAGGATAGCAGAGAA-3′
*R62G* Gγ_2_	5′-CTC**GCGGCCGC**TTAAAGGATAGCACAGAAAAACTTCTTCTCCCCAAA-3′
*K64G* Gγ_2_	5′-CTC**GCGGCCGC**TTAAAGGATAGCACAGAAAAACTTCCCCTCCCTAAA-3′
*K65G* Gγ_2_	5′-CTC**GCGGCCGC**TTAAAGGATAGCACAGAAAAACCCCTTCTCCCTAAA-3′
*R62G K64G* Gγ_2_	5′-CTC**GCGGCCGC**TTAAAGGATAGCACAGAAAAACTTCCCCTCCCCAAA-3′
*R62G K65G* Gγ_2_	5′-CTC**GCGGCCGC**TTAAAGGATAGCACAGAAAAACCCCTTCTCCCCAAA-3′
*K64G K65G* Gγ_2_	5′-CTC**GCGGCCGC**TTAAAGGATAGCACAGAAAAACCCCCCCTCCCTAAA-3′
*R62G K64G K65G* Gγ_2_	5′-CTC**GCGGCCGC**TTAAAGGATAGCACAGAAAAACCCCCCCTCCCCAAA-3′
*GER-R62G* Gγ_2_	5′-CTC**GCGGCCGC**TTAAAGGATAGCAGAGAAAAACTTCTTCTCCCCAAA-3′
*GER-K64G* Gγ_2_	5′-CTC**GCGGCCG**CTTAAAGGATAGCAGAGAAAAACTTCCCCTCCCTAAA-3′
*GER-K65G* Gγ_2_	5′-CTC**GCGGCCGC**TTAAAGGATAGCAGAGAAAAACCCCTTCTCCCTAAA-3′
*GER-R62G K64G* Gγ_2_	5′-CTC**GCGGCCGC**TTAAAGGATAGCAGAGAAAAACTTCCCCTCCCCAAA-3′
*GER-R62G K65G* Gγ_2_	5′-CTC**GCGGCCGC**TTAAAGGATAGCAGAGAAAAACCCCTTCTCCCCAAA-3′
*GER-K64G K65G* Gγ_2_	5′-CTC**GCGGCCG**CTTAAAGGATAGCAGAGAAAAACCCCCCCTCCCTAAA-3′
*GER-R62G K64G K65G* Gγ_2_	5′-CTC**GCGGCCGC**TTAAAGGATAGCAGAGAAAAACCCCCCCTCCCCAAA-3′

**Table 3 biomedicines-11-00557-t003:** Multiple sequence alignment of human Gγ proteins.

-MPVINIEDLTEKDKLKMEVDQLKKEVTLERMLVSKCCEEVRDYVEERSGEDPLVKGIPEDKN**PFKELKGGCVIS**	** G ** γ1
----MASN–NTASIAQARKLVEQLKMEANIDRIKVSKAAADLMAYCEAHAKEDPLLTPVPASEN**PFREKKFFCAIL**	** G ** γ ** 2 **
MKGETPVN–STMSIGQARKMVEQLKIEASLCRIKVSKAAADLMTYCDAHACEDPLITPVPTSEN**PFREKKFFCALL**	** G ** γ3
KEGMSNN–STTSISQARKAVEQLKMEACMDRVKVSQAAADLLAYCEAHVREDPLIIPVPASEN**PFREKKFFCTIL**	** G ** γ4
----MS----GSSSVAAMKKVVQQLRLEAGLNRVKVSQAAADLKQFCLQNAQHDPLLTGVSSSTN**PFRPQKV-CSFL**	** G ** γ5
----MS----ATNNIAQARKLVEQLRIEAGIERIKVSKAASDLMSYCEQHARNDPLLVGVPASEN**PFKDKKP-CIIL**	** G ** γ6
----MS-N–NMAKIAEARKTVEQLKLEVNIDRMKVSQAAAELLAFCETHAKDDPLVTPVPAAEN**PFRDKRLFCVLL**	** G ** γ7
-------MAQDLSEKDLLKMEVEQLKKEVKNTRIPISKAGKEIKEYVEAQAGNDPFLKGIPEDKN**PFKE-KGGCLIS**	** G ** γ8
----MS----SGASASALQRLVEQLKLEAGVERIKVSQAAAELQQYCMQNACKDALLVGVPAGSN**PFREPRS-CALL**	** G ** γ9
---MPALHIEDLPEKEKLKMEVEQLRKEVKLQRQQVSKCSEEIKNYIEERSGEDPLVKGIPEDKN**PFKE-KGSCVIS**	** G ** γ10
----MSSKTASTNNIAQARRTVQQLRLEASIERIKVSKASADLMSYCEEHARSDPLLIGIPTSEN**PFKDKKT-CIIL**	** G ** γ12
---------MEEWDVPQMKKEVESLKYQLAFQREMASKTIPELLKWIEDGIPKDPFLNPDLMKNN**PWVE-KGKCTIL**	** G ** γ13

**Table 4 biomedicines-11-00557-t004:** Site-directed mutagenesis of Gγ_2_.

*WT*	PLLTPVPA*S*ENPF R E KK FF C AIL	PLLTPVPA*S*ENPF R E KK FF *S* AIL	*ger-*
*K65*	PLLTPVPA*S*ENPF R E K *G* FF C AIL	PLLTPVPA*S*ENPF R E K *G* FF *S* AIL	*ger-K65*
*K64*	PLLTPVPA*S*ENPF R E *G* K FF C AIL	PLLTPVPA*S*ENPF R E *G* K FF *S* AIL	*ger-K64*
*R62*	PLLTPVPA*S*ENPF *G* E KK FF C AIL	PLLTPVPA*S*ENPF *G* E KK FF *S* AIL	*ger-R62*
*K64K65*	PLLTPVPA*S*ENPF R E *GG* FF C AIL	PLLTPVPA*S*ENPF R E *GG* FF *S* AIL	*ger-K64K65*
*R62K65*	PLLTPVPA*S*ENPF *G* E K *G* FF C AIL	PLLTPVPA*S*ENPF *G* E K *G* FF *S* AIL	*ger-R62K65*
*R62K64*	PLLTPVPA*S*ENPF *G* E *G* K FF C AIL	PLLTPVPA*S*ENPF *G* E *G* K FF *S* AIL	*ger-R62K64*
*R62K64K65*	PLLTPVPA*S*ENPF *G* E *GG* FF C AIL	PLLTPVPA*S*ENPF *G* E *GG* FF *S* AIL	*ger-R62K64K65*

## Data Availability

Not applicable.

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
