# Peer review of "Structural Basis of the Interaction of the G Proteins, Gαi1, Gβ1γ2 and Gαi1β1γ2, with Membrane Microdomains and Their Relationship to Cell Localization and Activity"

_biomedicines, 2023, doi:10.3390/biomedicines11020557_

Round 1

Reviewer 1 Report

This is a nice and well written paper presented by authors with a large experience in the field. The overall goal of the study is to investigate the role of membrane lipid structure, along with G-proteins lipid modifications and charge, on the binding of monomeric, dimeric and trimeric G proteins to lipid membranes. These interactions are critical for cell signal propagation, and their alterations are involved in pathological processes. Besides its interest in terms of basic knowledge, the authors very cleverly use the accumulated knowledge to develop potential new therapeutic strategies, termed melitherapy, based on using synthetic lipids to modify such interactions.

I should also indicate that I find this paper particularly suitable to be included in a special issue to celebrate the 50th aniversary of the fluid mosaic model of biological membranes, as it comes to emphasize the current view of the fluid mosaic model in terms of a dynamically-structured patchwork of membrane microdomains. Such protein-lipid microdomains are in continuous dynamic turnover, resulting in activation or deactivation of “lipid switches” that affect crucial cellular processes such as differentiation, cell proliferation, cell death, etc.

I have no major concerns regarding this paper. Only a very minor issue to point out: the word "in" is missing from ...is involved in many cellular processes.. (line 54).

Author Response

REVIEWER 1

Comments and Suggestions for Authors

This is a nice and well written paper presented by authors with a large experience in the field. The overall goal of the study is to investigate the role of membrane lipid structure, along with G-proteins lipid modifications and charge, on the binding of monomeric, dimeric and trimeric G proteins to lipid membranes. These interactions are critical for cell signal propagation, and their alterations are involved in pathological processes. Besides its interest in terms of basic knowledge, the authors very cleverly use the accumulated knowledge to develop potential new therapeutic strategies, termed melitherapy, based on using synthetic lipids to modify such interactions.

I should also indicate that I find this paper particularly suitable to be included in a special issue to celebrate the 50th aniversary of the fluid mosaic model of biological membranes, as it comes to emphasize the current view of the fluid mosaic model in terms of a dynamically-structured patchwork of membrane microdomains. Such protein-lipid microdomains are in continuous dynamic turnover, resulting in activation or deactivation of “lipid switches” that affect crucial cellular processes such as differentiation, cell proliferation, cell death, etc.

I have no major concerns regarding this paper. Only a very minor issue to point out: the word "in" is missing from ...is involved in many cellular processes.. (line 54).

RESPONSE

We thank the reviewer’s comments. The indicated change has been addressed and “in” was correctly placed in the sentence (line 54).

Reviewer 2 Report

To Authors

In the present study, this study has identified lipid modifications of the α subunit and the g subunit of G protein and their binding to specific lipid domains. This study demonstrated (1) that binding to PE is governed by differences in lipid and C-terminal basic amino acids of the g subunit, and (2) that C-terminal basic amino acids govern binding to PS, and (3) that binding of the Gαβg trimer to PE-rich membranes is also governed by the α subunit. Furthermore, the binding of the Gαβg trimer to PE-rich membranes requires lipid modification of the α subunit and geranylgeranylation of the g subunit. These results support the previously assumed idea. The points of concern are discussed below.

Major Comments.

1. The conclusion supports the previously thought idea that the lipid modification of the α subunit and g subunit as well as the polybasic domain of the g subunit is a key to membrane localization. New finding or approach are preferred to be included.

2. Can the present results explain the differences in the localization of various G proteins?

3. It is inferred from the present results that G proteins undergoing the same lipid modification are expressed in the same region or domain, whether they are Gs, Gi, Gq, or G12/13 family G proteins. It would be better to discuss whether this idea is consistent with previous data.

4. Figures 9B and 9D, when differences in palmitoylation and myristoylation of the α subunit promote binding to PE-rich domains and modification of the g subunit promotes binding to PS/PE-rich domains, it is recommended that interactions between lipids and α or g subunits should be shown by FRET, BRET, or other methods. This should further support the present results.

5. In geranylgeranylation, the terminal 3 amino acids are removed, and the new terminal amino acid is methylated. The effects of unmethylation are not examined in this study.

Minor comments.

1. It is recommended to make a table of the selectivity of lipid binding (e.g., PE>PC) for the mutants of the α and g subunits, as this will help us to better understand the results.

2. The amount of G protein bound to lipids is detected with antibodies. It would be better to include figures showing a linear relationship between the amount of G protein detected and the amount of signal generated.

3. Vertical axis of Figure 1A~D, the explanatory text of the figure says that it is shown as a ratio of wild-type Gβ, but it would be better to put the amount of original Gβ bound in each graph. If other figures are also expressed as ratios, the corresponding wild-type binding levels should also be included.

Author Response

REVIEWER 2

Comments and Suggestions for Authors

To Authors

In the present study, this study has identified lipid modifications of the α subunit and the g subunit of G protein and their binding to specific lipid domains. This study demonstrated (1) that binding to PE is governed by differences in lipid and C-terminal basic amino acids of the g subunit, and (2) that C-terminal basic amino acids govern binding to PS, and (3) that binding of the Gαβg trimer to PE-rich membranes is also governed by the α subunit. Furthermore, the binding of the Gαβg trimer to PE-rich membranes requires lipid modification of the α subunit and geranylgeranylation of the g subunit. These results support the previously assumed idea. The points of concern are discussed below.

We thank the comments of the reviewer. Response to his or her questions will be indicated in boldface below for easy follow up.

Major Comments.

  1. The conclusion supports the previously thought idea that the lipid modification of the α subunit and g subunit as well as the polybasic domain of the g subunit is a key to membrane localization. New finding or approach are preferred to be included.

We appreciate the comment from this reviewer. It is correct that the present results constitute a new finding that explains the localization of G proteins on the basis of their lipid modifications and polybasic domain, as it was suggested previously. We have included, as suggested by the reviewer, the consideration of new finding in the Abstract (13th line) and Discussion sections (lines 665-668).

  1. Can the present results explain the differences in the localization of various G proteins?

This is a very interesting question. As the different G protein subunits have different lipid modifications in the alpha and gamma subunits and their amino acid sequence differ in the polybasic domain region, it could be expected that their localizations would vary according to their structure and combinations of the alpha, beta and gamma subunits. This point has been highlighted in the Discussion section (lines 885-888).

  1. It is inferred from the present results that G proteins undergoing the same lipid modification are expressed in the same region or domain, whether they are Gs, Gi, Gq, or G12/13 family G proteins. It would be better to discuss whether this idea is consistent with previous data.

The reviewer is correct in his or her appreciation. The present results would explain that similar regions or domains have similar localizations. This has been reflected in the conclusion section (lines 906-908) and a new reference [76] was added to discuss previous existing data suggesting this point.

  1. Figures 9B and 9D, when differences in palmitoylation and myristoylation of the α subunit promote binding to PE-rich domains and modification of the g subunit promotes binding to PS/PE-rich domains, it is recommended that interactions between lipids and α or g subunits should be shown by FRET, BRET, or other methods. This should further support the present results.

Previous studies from our group using confocal microscopy (reference 6 in the manuscript) and other spectroscopic techniques (reference 43 in the manuscript) suggest the differential interaction of the Ggamma subunit with different amino acids or lipids in the C-terminal region. This point has been indicated in lines 684-685. However, the comment by the reviewer is very interesting and goes beyond the above-mentioned references. Thus, we have been working on the co-localization of G proteins and membrane lipids during the last months. A full study on this co-localization will be prepared with these data.

  1. In geranylgeranylation, the terminal 3 amino acids are removed, and the new terminal amino acid is methylated. The effects of unmethylation are not examined in this study.

Mutation of the C-terminus of Gg2 protein (C68S) has a double effect on the protein modification, as it induces a lack of (1) isoprenylation, (2) methylation, and (3) C-terminal proteolysis. This point has been highlighted in lines 685-688, in addition to lines 316-319, which already indicated this point in the previous version.

Minor comments.

  1. It is recommended to make a table of the selectivity of lipid binding (e.g., PE>PC) for the mutants of the α and g subunits, as this will help us to better understand the results.

This is a very complex task as the affinity of the mutants in different forms of the G proteins (i.e., the monomer, dimer and trimer: Ga1, Gb1g2, Ga1b1g2) is not the same. In addition, the effect of a single-point mutant, which reflects the role of the amino acid residue, can be modified by an additional mutation. Therefore, the lipid preference of proteins with mutated amino acids would have to be indicated in the context of the different G protein forms and mutation combinations evaluated. We believe that more studies using different subunits and mutations would shed light on this relevant point highlighted by the reviewer and might be the basis for a Review paper in which a discussion of all the forms investigated would be addressed.

  1. The amount of G protein bound to lipids is detected with antibodies. It would be better to include figures showing a linear relationship between the amount of G protein detected and the amount of signal generated.

The measurement of protein amounts using immunoblotting is similar in its implementation to the determination of protein levels in solution using a standard curve (e.g., the Lowry or Bradford Methods). This method of measurement is described in Escribá et al., Arch. Gen. Psychiatry 1994; 51:494-501. As the number of references in the present study was high (76 references), we decided not to reference these methodological papers, already described in many papers from our and other research groups.

  1. Vertical axis of Figure 1A~D, the explanatory text of the figure says that it is shown as a ratio of wild-type Gβ, but it would be better to put the amount of original Gβ bound in each graph. If other figures are also expressed as ratios, the corresponding wild-type binding levels should also be included.

This point has been addressed and the legend in Figure 1 indicates the ng of Gb bound per ng of Gb in the medium.